# Flagellar rotation facilitates the transfer of a bacterial conjugative plasmid

Saurabh Bhattacharya[1,2], Michal Bejerano-Sagie [1,2], Miriam Ravins[1], Liat Zeroni [1], Prabhjot Kaur[1], Venkadesaperumal Gopu[1], Ilan Rosenshine [1✉] & Sigal Ben-Yehuda [1✉]

## Abstract

**Conjugation-mediated DNA delivery is the primary mode for antibiotic resistance spread in bacteria; yet, molecular mechanisms regulating the conjugation process remain largely unexplored. While conjugative plasmids typically require bacterial attachment to solid surfaces for facilitation of donor-to-recipient proximity, the pLS20 conjugative plasmid, prevalent among Gram-positive *Bacillus* spp., uniquely requires fluid environments to enhance its transfer. Here, we show that pLS20, carried by *Bacillus subtilis*, induces multicellular clustering, which can accommodate various species, hence offering a stable platform for DNA delivery in a liquid milieu. We further discovered that induction of pLS20 promoters, governing crucial conjugative genes, is dependent on the presence of donor cell flagella, the major bacterial motility organelle. Moreover, the pLS20 regulatory circuit is controlled by a mechanosensing signal transduction pathway responsive to flagella rotation, thus activating conjugation gene expression exclusively during the host motile phase. This flagella-conjugation coupling strategy may allow the dissemination of the plasmid to remote destinations, allowing infiltration into new niches.**

**Keywords** *Bacillus subtilis*; Conjugation; Flagella; Mating Pair Formation; Motility
**Subject Categories** DNA Replication, Recombination & Repair; Microbiology, Virology & Host Pathogen Interaction; Signal Transduction
Published online: 2 Decembre 2024

## Introduction

Conjugation, discovered by Lederberg and Tatum in 1946, is a key mechanism of horizontal gene transfer within natural bacterial communities, allowing a unidirectional transfer of genetic material from a donor to a recipient in a contact-dependent manner (Lederberg and Tatum, 1946). Conjugative plasmids, carrying genetic information essential for their autonomous transfer, play a central role in the acquisition and dissemination of antibiotic

resistance and virulence factors, thereby contributing to bacterial endurance and pathogenesis (e.g., Partridge et al, 2018; Ochman et al, 2000; Grohmann et al, 2003). Successful conjugative transmission necessitates a tight association between donor and recipient bacteria, known as mating pair formation (e.g., Thomas and Nielsen, 2005; Low et al, 2022). While Gram-negative conjugation systems typically achieve this intimate interaction through extracellular filaments, designated pili (e.g., Babić et al, 2008; Goldlust et al, 2023; Costa et al, 2016; Cabezón et al, 2014), the mechanisms underlying mating pair formation in Gram-positives remain largely elusive, as their conjugation systems appear to lack a pilus or a similar structure (e.g., Grohmann et al, 2003). Nonetheless, fundamental aspects of conjugation are conserved across Gram-positive and -negative bacteria. Both mechanisms are typified by converting conjugative plasmid DNA from double-stranded into a single-stranded form, a process driven by a plasmid-encoded relaxase. The resulting single-stranded DNA (ssDNA), protected by plasmid-encoded ssDNA binding protein (SSB), is directed to the transferosome, a membrane-associated mating complex housing a type IV secretion system (T4SS). Through this apparatus, the plasmid ssDNA is delivered to a recipient cell (transconjugant), providing a template for generating a double-stranded plasmid (e.g., Grohmann et al, 2003; Cabezón et al, 2014; Waksman, 2019; Goessweiner-Mohr et al, 2014; Ilangovan et al, 2017).

The 65 Kb conjugative plasmid pLS20, initially isolated from the Gram-positive *Bacillus subtilis* (*B. subtilis*) *natto* strain (Tanaka et al, 1977), possesses transmission capacity to other *Bacilli*, including the pathogenic species *B. anthracis* and *B. cereus*, highlighting its potential to serve as a vehicle for interspecies genetic exchange (Koehler and Thorne, 1987; Itaya et al, 2022; Mori et al, 2022). In fact, pLS20 serves as the archetype plasmid to a family of at least 35 members, dispersed across various *Bacillus* species residing in diverse ecological niches (Val-Calvo et al, 2021a). It carries a gene cluster encoding the T4SS machinery, SSB, relaxase, as well as additional essential conjugation functions (Ramachandran et al, 2017; Miguel-Arribas et al, 2017; Val-Calvo et al, 2021b; Crespo et al, 2022). The primary conjugation promoter $P_C$ is repressed by Rco, maintaining the default "OFF" state of conjugation. In parallel, an anti-repressor, termed Rap, is held inactive through binding to a short quorum-sensing signaling

[1]Department of Microbiology and Molecular Genetics, Institute for Medical Research Israel-Canada (IMRIC), The Hebrew University-Hadassah Medical School, POB 12272, The Hebrew University of Jerusalem, 91120 Jerusalem, Israel. [2]These authors contributed equally: Saurabh Bhattacharya, Michal Bejerano-Sagie. ✉E-mail: ilanr@ekmd.huji.ac.il; sigalb@ekmd.huji.ac.il

peptide, Phr, encoded by the plasmid. The transition to an "ON" state is facilitated by a decrease in Phr concentration, and the consequent liberation of Rap, which counteracts Rco, thus promoting conjugation (Singh et al, 2013; Ramachandran et al, 2014; Bernardo et al, 2023; Mori et al, 2021).

Intriguingly, unlike most known conjugative systems that rely on a solid surface for efficient DNA delivery, pLS20 conjugation is typically executed in liquid medium, when the bacterium is highly motile (Itaya et al, 2006). Notably, *B. subtilis* exhibits two distinct lifestyles: a motile mode dominated by flagellated cells and a sessile mode marked by long cell chains, often devoid of flagella (Kearns and Losick, 2005; Diethmaier et al, 2011). The switch between these modes is governed, among others, by the phosphodiesterase YmdB, with cells lacking *ymdB* displaying increased flagellation but are deficient in forming biofilms and intercellular membranous nanotube bridges (Diethmaier et al, 2011, 2014; Dubey et al, 2016). Flagella biogenesis is initiated through the assembly of the flagellar type III secretion system, encoded by *fliOPQR flhBA* genes, collectively termed *CORE* (Bhattacharya et al, 2019; Minamino, 2014; Mukherjee and Kearns, 2014; Diepold and Armitage, 2015). This assembly, along with subsequent basal body and hook formation, triggers the secretion of FlgM, an anti-sigma factor inhibiting the activity of the flagellar sigma factor, SigD. Upon release of SigD from FlgM inhibition, the expression of various flagellar genes, including *hag*, encoding the major filament subunit, is induced, leading to the formation of the flagellar filament on the cell surface (Mukherjee and Kearns, 2014; Serizawa et al, 2004; Calvo and Kearns, 2015; Caramori et al, 1996; Phillips et al, 2015).

Here we explore the enigmatic connection between *B. subtilis* motility and conjugation. We reveal that flagella are essential for effective conjugation in liquid media, by serving as a mechanosensory device prompting the transcription of genes essential for pLS20 conjugation in a subset of cells. Successively, these donor cells acquire the ability to organize mating clusters, fostering proximity between donor and recipient bacteria, establishing a robust foundation for mating pair formation, and ultimately facilitating fruitful plasmid transfer.

# Results

## Visualization of conjugation revealed the formation of mating clusters

The dependency of pLS20 conjugation on *B. subtilis* motile lifestyle (Itaya et al, 2006), prompted us to explore how donor and recipient cells are coupled to fasten plasmid delivery in fluid surroundings. To track conjugation events at a single-cell resolution, we initially employed pLS20, modified to include *gfp* driven by $P_{IPTG}$ promoter, though GFP expression kinetics in the transconjugants was relatively slow (Appendix Fig. S1A,B). To improve transconjugant detection, the native pLS20-*ssb* gene, encoding ssDNA binding protein (Singh et al, 2013), one of the earliest proteins to be robustly expressed in transconjugants (Masai and Arai, 1997; Couturier et al, 2023), was fused to YFP. The SSB-YFP fusion was fully functional in supporting conjugation (Appendix Fig. S1C), and was manifested within donor bacteria as faint foci (Fig. 1A). When donor cells carrying pLS20-*ssb-yfp* were mixed with mCherry-labeled recipients, bright prominent SSB-YFP foci subsequently emerged in transconjugant bacteria, as early as 60 min post mixing (Fig. 1A; Appendix Fig. S1B). The appearance of these foci reported conjugation events wherein pLS20 was successfully delivered to recipient bacteria, leading to the production of SSB-YFP.

Consequently, we developed a microscopy-based conjugation assay under semi-liquid conditions (0.6% agarose), allowing to follow the kinetics of conjugation events by time-lapse microscopy. Intriguingly, promptly after mixing donor and recipient bacteria in a 1:1 ratio, cell clusters emerged, comprising a few donors surrounded by numerous recipients. These cell assemblages remained stable and expanded over time both through cell division and the incorporation of additional

**Figure 1. MC formation is promoted by pLS20 and donor flagella.**

(A) Donor cells (dark) (SH347: WT/pLS20$_{cm}$-*ssb-yfp*) were mixed with recipient cells (red) (BDR2637: *sacA*::P$_{veg}$-*mCherry*) in 1:1 ratio (T = 0 min), incubated for 60 min and visualized by fluorescence microscopy. Images were set to co-visualize faint (donor) and bright (transconjugant) SSB-YFP foci. Left panel: Shown is a representative overlay image of phase contrast (gray), with fluorescence from mCherry (red) and SSB-YFP (yellow) captured at 0 min. Right panels: Shown are representative images of fluorescence from (left to right): SSB-YFP (yellow), an overlay of phase contrast (gray) with fluorescence from SSB-YFP (yellow), and an overlay of phase contrast (gray), with fluorescence from mCherry (red) and SSB-YFP (yellow). White arrows highlight donor cells expressing faint SSB-YFP foci. Cyan arrows highlight transconjugants expressing bright SSB-YFP foci. A representative experiment out of three independent biological repeats. (B) Donor cells (dark) (SH347: WT/pLS20$_{cm}$-*ssb-yfp*) were mixed with recipient cells (red) (BDR2637: *sacA*::P$_{veg}$-*mCherry*) in 1:1 ratio, placed over semi-solid (0.6%) LB agarose pad, and visualized by time-lapse fluorescence microscopy. Upper panels: overlay images of phase contrast (gray) with fluorescence from mCherry (red) and SSB-YFP (yellow), captured at the indicated time points. Lower panels: Schematics depicting cell layout of the corresponding upper panels. Arrows highlight a transconjugant cell. (C) Donor cells (dark) (SH337: WT/pLS20$_{cm}$) were mixed with recipient cells (red) (BDR2637: *sacA*::P$_{veg}$-*mCherry*) in 1:1 ratio, and the formation of MCs in liquid medium was followed using digital wide-field microscopy. A strain lacking pLS20$_{cm}$ (PY79: WT) was used as a control. Shown are representative images captured at the indicated time points after mixing: phase contrast (gray), fluorescence from mCherry-labeled recipients (red), computed clustering analysis (outlines), and overlay of phase contrast and mCherry fluorescence. Clustering analysis was derived from images of mCherry channel subjected to thresholding and analyzed for particles larger than 500 pixels$^2$. A representative experiment out of 3 independent biological repeats. (D) Quantification of the clustering analysis shown in (C). The presence or absence of pLS20$_{cm}$ is indicated. For each mixture, triplicates were analyzed in parallel, and total area of clusters larger than 500 pixels$^2$ was calculated. Data is presented as average values and SEM. Statistical significance was calculated using two-way ANOVA. P values: (*) = 2.7 × 10$^{-2}$. ND not detected. (E) *B. subtilis* (*Bs*) donor cells (green) (SH501: *sacA*::P$_{33}$-*gfp*/pLS20$_{cm}$) were mixed with *B. subtilis* recipient cells (red) (BDR2637: *sacA*::P$_{veg}$-*mCherry*) and *B. cereus* (*Bc*) (dark) (left panel) or *B. megaterium* (*Bm*) (dark) (right panel) cells in 1:1:1 ratio, placed over semi-solid (0.6%) LB agarose pad for 45 min and visualized by fluorescence microscopy. Shown are overlay images of phase contrast (gray), with fluorescence from mCherry (red) and GFP (green). (F) Donor strains (dark): WT (SH337), Δ*hag* (SH443), or Δ*flgM* (SH496) harboring pLS20$_{cm}$, were mixed with recipient cells (red) (BDR2637: *sacA*::P$_{veg}$-*mCherry*) in 1:1 ratio, and the formation of MCs in liquid medium was followed using digital wide-field microscopy. Shown is quantification of the clustering analysis at the indicated time points after mixing. Representative images for time point 30 min are displayed in Appendix Fig. S2A. For each mixture, triplicates were analyzed in parallel, and total area of clusters larger than 500 pixels$^2$ was calculated. Data are presented as average values and SEM. Statistical significance between WT and each mutant was calculated using two-way ANOVA. P values: (ns not significant) = 4.8 × 10$^{-1}$, (**) = 4.5 × 10$^{-3}$. A representative experiment out of three independent biological repeats. Source data are available online for this figure.

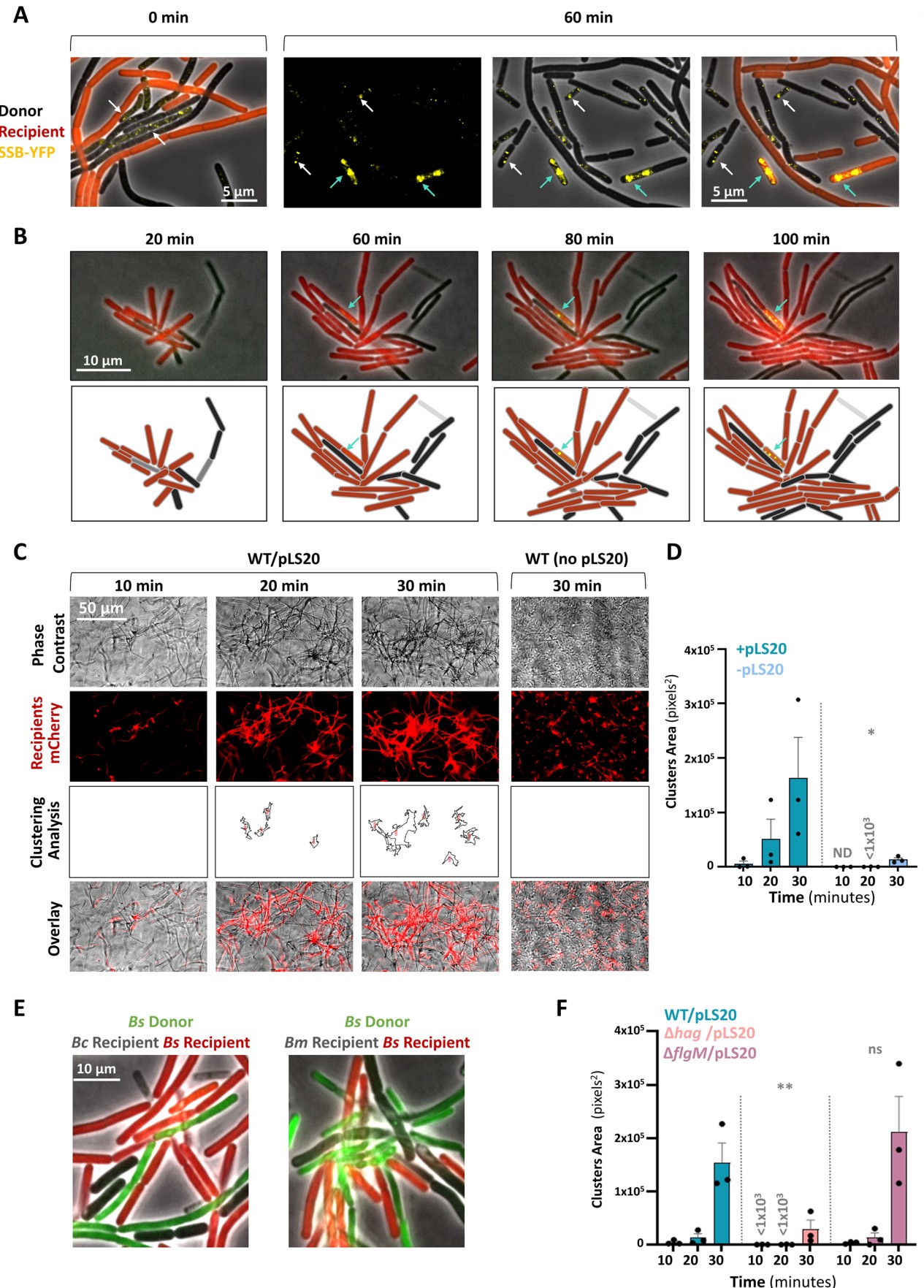

bacteria (Fig. 1B). Approximately 60–80 min post co-incubation, recipient transconjugants expressing SSB-YFP appeared adjacent to donor bacteria; hence, reporting effective conjugation events (Fig. 1B). Notably, transconjugants were predominantly cluster-associated (~85%) (Appendix Fig. S1D), suggesting that donor cells induce the formation of multicellular "mating-clusters" (MCs), providing a platform for enhancing conjugation. To further examine this notion, we designed an assay for quantitative monitoring of MC formation kinetics after mixing donors with mCherry-labeled recipient cells, using digital wide-field microscopy. This allowed the visualization of MC formation under liquid conditions in large fields. MCs, dominated by recipient bacteria, were typically evident 20 min post co-incubation, becoming more pronounced over time (Fig. 1C,D). Importantly, MC development was pLS20-dependent, strongly linking the phenomenon to the conjugation process (Fig. 1C,D). Since pLS20-like plasmids are prevalent in *Bacilli* (Val-Calvo et al, 2021a), we investigated whether species other than *B. subtilis* could be recruited to pLS20-derived MCs. Therefore, MC formation was visualized within mixed populations comprising differentially-labeled *B. subtilis* donor and recipient cells, alongside unlabeled *B. cereus* or *B. megaterium* cells. Remarkably, cells from both species were observed associating with *B. subtilis* MCs, and were often adjacent to potential *B. subtilis* donors (Fig. 1E).

Taken together, during conjugation, pLS20-induced MCs are assembled, containing a few donors outnumbered by potential recipients, which could belong to the same or different species. These MCs appear to provide a relatively static hub for promoting intra- and inter- species mating pair formation in liquid conditions.

## Donor-produced flagella are required for conjugation

To address whether bacterial motility impacts MC establishment, a mutant lacking the flagellum filament (Δ*hag*) and a hyper-flagellated mutant, overexpressing the flagellar genes (Δ*flgM*)

(Phillips et al, 2015; Mukherjee and Kearns, 2014), both carrying pLS20, were tested for their ability to assemble MCs. Surprisingly, the kinetics of MC formation was largely impeded in the non-motile Δ*hag* donor, whereas the hyper-motile Δ*flgM* showed enhanced clustering capacity (Fig. 1F; Appendix Fig. S2A,B). Nonetheless, the lack of flagella in the recipient did not perturb MC formation (Appendix Fig. S2C), excluding that flagella entangling or adhesion properties (Haiko and Westerlund-Wikström, 2013; Friedlander et al, 2015; Sjoblad et al, 1985) are involved in this process. We then explored whether the dependency of MC formation on flagella correlates with respective conjugation efficiencies, utilizing the SSB-YFP reporter. No discernible conjugation events were monitored for Δ*hag* donor, whereas the Δ*flgM* donor yielded elevated transconjugant levels compared to wild-type (WT) (Fig. 2A,B). These results indicate that MC formation and the subsequent pLS20 conjugation are promoted by donor flagella.

To substantiate the link between the presence of flagella and conjugation, we quantified the rate of emergence of transconjugant colony-forming units (CFUs), in various donor flagellar mutants, concomitantly with flagella visualization and motility (Fig. 2C–E). In agreement with the microscopy-based assay, the non-motile Δ*hag* cells were severely conjugation deficient (Fig. 2E). We further tested additional non-motile mutants perturbed in flagella biogenesis: Δ*CORE* deleted for the genes encoding the flagellar export apparatus, and Δ*sigD*, lacking the flagellar sigma factor (Mukherjee and Kearns, 2014; Serizawa et al, 2004) (Fig. 2C,D). Both mutants were strongly attenuated in pLS20 conjugation (Fig. 2E). Moreover, in line with the microscopy findings, the hyper-motile mutant, Δ*flgM*, exhibited increased conjugation levels, that were further boosted in Δ*ymdB* mutant overexpressing flagella (Diethmaier et al, 2011) (Fig. 2C–E; Appendix Fig. S2B). Interestingly, a non-motile mutant lacking *motA*, which assembles paralyzed flagella

**Figure 2. Functional flagella facilitate conjugation.**

(A) Donor strains (dark): WT (SH347), Δ*hag* (MBS20), or Δ*flgM* (SH561), harboring pLS20$_{cm}$-*ssb-yfp*, were mixed with recipient cells (red) (BDR2637: *sacA*::P$_{veg}$-*mCherry*) in 1:1 ratio, incubated for 60 min and visualized by fluorescence microscopy. Shown are overlay images of phase contrast (gray), with fluorescence from mCherry (red) and SSB-YFP (yellow) of the indicated strains. Arrows highlight transconjugant cells. (B) Conjugation efficiencies of the donors described in (A) were calculated as the number of transconjugants [mCherry (red) cells displaying SSB-YFP (yellow) foci] (T)/number of total recipients [mCherry (red) cells] (T + R). Plotted are conjugation efficiencies derived from a representative experiment out of 3 independent biological repeats. $n > 3000$ cells for each strain. ND—not detected. (C) The following strains: WT (DS1895), Δ*CORE* (SH411), Δ*sigD* (SH415), Δ*motA* (SH409), Δ*flgM* (SH408), and Δ*ymdB* (SH101), harboring modified flagellin *hag*$^{T209C}$ (*amyE*::P$_{hag}$-*hag*$^{T209C}$), were grown in liquid LB, flagellin was stained with Alexa Fluor 594 C$_5$ maleimide, and cells were visualized by fluorescence microscopy. Shown are images of phase contrast (gray) (upper panels) and corresponding stained flagella (red) (lower panels). (D) The following strains: WT (PY79), Δ*CORE* (SH9), Δ*sigD* (IB97), Δ*motA* (SH419), Δ*flgM* (SH495), and Δ*ymdB* (GB61) were grown to mid-logarithmic phase and subjected to motility assay by spotting onto LB plates supplemented with 0.3% agar. Shown are images captured after 7 h of incubation. (E) Donor strains: WT (SH337), Δ*hag* (PK143), Δ*CORE* (SH352), Δ*sigD* (PK143), Δ*motA* (SH423), Δ*flgM* (SH496), or Δ*ymdB* (SH442) harboring pLS20$_{cm}$ were mixed with recipient cells (SH345: *sacA*::*kan* or SH360: *sacA*::*spec*) in 1:1 ratio, incubated for 20 min, and serial dilutions were spotted either on LB agar containing chloramphenicol and kanamycin or chloramphenicol and spectinomycin, selecting for transconjugants. In parallel, dilutions were spotted on either kanamycin or spectinomycin-containing plates, selecting for recipients. Upper panel: Conjugation efficiencies calculated as the number of transconjugants (T)/ number of total recipients (T + R). For each strain, at least three independent biological repeats were conducted. Data are shown as box plot graphs. The box is determined by the 25th and 75th percentiles, and whiskers are determined by min and max; the line in the box indicates the median. Statistical significance between WT and each mutant was calculated using Student's *t* test. *P* values: Δ*hag* (***) = $1.82 \times 10^{-5}$, Δ*CORE* (****) = $2.16 \times 10^{-5}$, Δ*sigD* (****) = $7.34 \times 10^{-5}$, Δ*motA* (***) = $2.68 \times 10^{-4}$, Δ*flgM* (*) = $2.85 \times 10^{-2}$, Δ*ymdB* (*) = $1.57 \times 10^{-2}$. Lower panels: Representative images of spotted conjugation mixtures ($10^{-2}$ dilution) over LB agar containing chloramphenicol and kanamycin or chloramphenicol and spectinomycin, selecting for transconjugants. Indicated conjugation efficiencies were calculated as % of WT conjugation efficiency. Presented are average values and SD of at least three independent biological repeats. (F) Donor cells (SH337: WT/pSL20$_{cm}$) were mixed with WT (SH345: *sacA*::*kan*) or Δ*hag* (MBS11: Δ*hag*, *amyE*::P$_{IPTG}$-*gfp-kan*) recipient cells in 1:1 ratio, incubated for 20 min, and serial dilutions were spotted either on LB agar containing chloramphenicol and kanamycin or solely kanamycin, selecting for transconjugants and recipients, respectively. Upper panel: Conjugation efficiencies calculated as the number of transconjugants (T)/number of total recipients (T + R). For each strain, at least three independent biological repeats were conducted. Data are shown as box plot graphs. The box is determined by the 25th and 75th percentiles, and whiskers are determined by min and max; the line in the box indicates the median. Statistical significance was calculated using Student's *t* tests. *P* values: (ns not significant) = $6.6 \times 10^{-1}$. Lower panels: Representative images of spotted conjugation mixtures ($10^{-2}$ dilution) over LB agar containing chloramphenicol and kanamycin, selecting for transconjugants. Indicated conjugation efficiencies were calculated as % of WT conjugation efficiency. Presented are average values and SD of at least four independent biological repeats. Source data are available online for this figure.

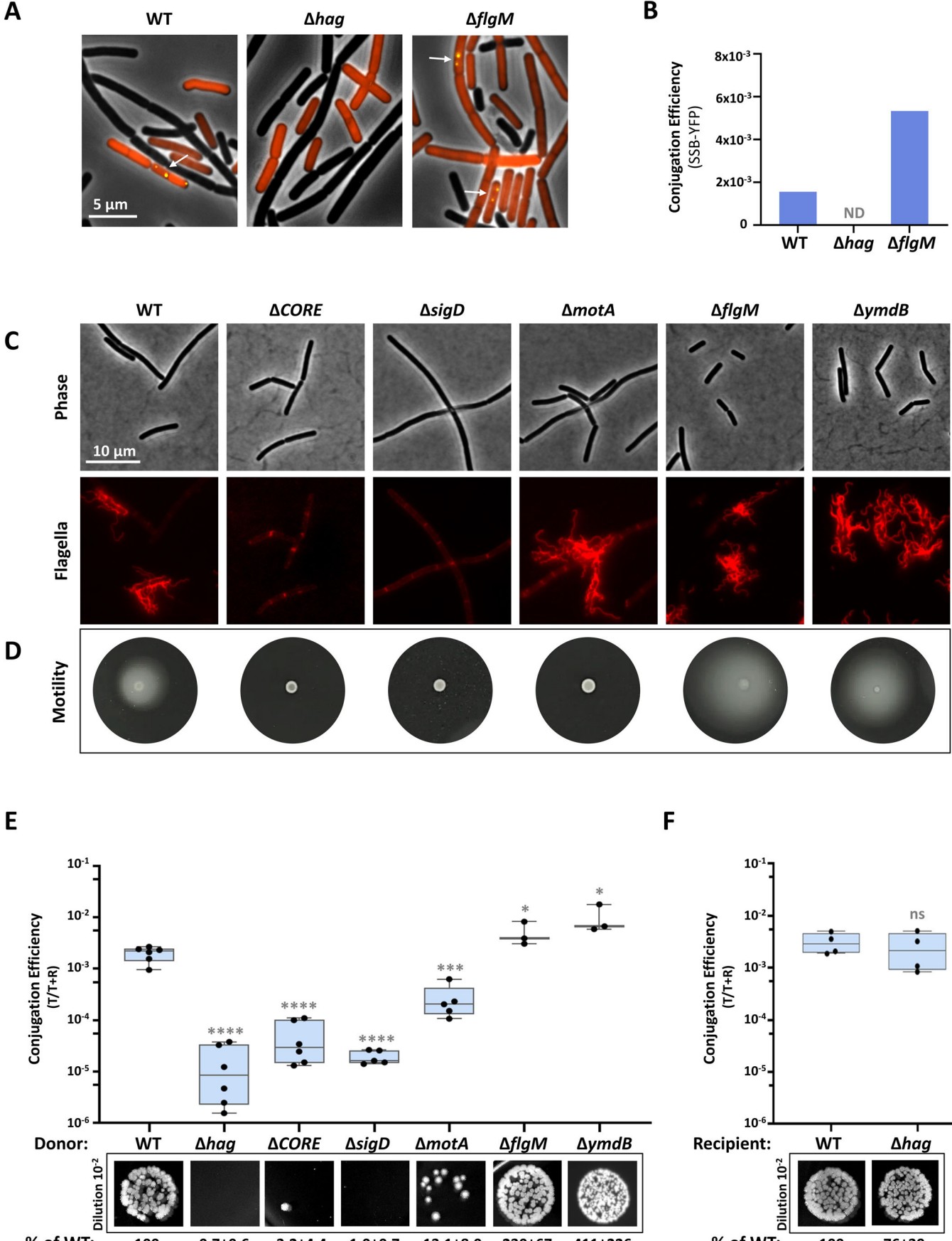

(Chevance and Hughes, 2008), showed decreased conjugation ability (Fig. 2C–E), suggesting that rotating flagella is required for the process.

To inspect whether the necessity of flagella for conjugation is solely a donor property or extends to the recipient, conjugation was conducted by mating a WT donor with either WT or Δ*hag* recipients. Similar numbers of transconjugants were obtained for both recipients, signifying that the need for flagella is a donor-specific demand (Fig. 2F), consistent with the flagella dispensability for recipient bacteria during MC formation (Appendix Fig. S2C). Collectively, our results indicate that functional rotating donor flagella collaborate with pLS20 factors to induce MC formation and conjugation occurrence.

## Flagella are essential for conjugation gene expression

To study the nature of the link between flagellum functionality and conjugation events, we assessed whether the necessity for flagella within donor cells could be bypassed by enhancing the transcription of conjugative genes. To achieve this, the anti-Rco conjugation repressor, Rap$_{pLS20}$ (herein RapA) (Singh et al, 2013; Mori et al, 2021), was ectopically overexpressed in flagellar mutants and assayed for conjugation. Indeed, ectopic expression of RapA effectively bypassed the conjugation defect in all tested flagellar mutants (Fig. 3A), hinting that flagella might play a role in augmenting the transcription of conjugation genes.

The potential flagella-conjugation gene expression axis was further examined by conducting quantitative Reverse Transcriptase PCR (qRT-PCR) on select pLS20 genes, comparing their transcription between WT and Δ*hag* mutant. A modest, yet significant, reduction in the RNA levels of *rapA*, along with ORFs 28 to 30, was detected in the Δ*hag* mutant (Fig. 3B,C). Furthermore, a sharper drop in the RNA amounts of ORF33 and onwards was evident, including transcripts of key conjugation genes such as *tie* (ORF34), encoding a putative adhesin, as well as *virB11* and *virD4*, encoding T4SS components (Singh et al, 2013; Gago-Córdoba et al, 2021)

(Fig. 3B,C). This gene expression dependency on flagella was emphasized by assessing *tie* transcription in various flagellar mutants, corroborating this phenomenon (Fig. 3D).

To strengthen these findings, we surveyed the protein level of crucial conjugation factors within flagellar mutants. pLS20-encoded *conAn1* (ORF28), an anti-termination factor (Miguel-Arribas et al, 2021), and *tie* were HA-tagged, and their production examined. ConAn1-$_{2xHA}$ levels were markedly reduced in the flagella-deficient mutants, while exhibiting increased synthesis in the hyper-flagellated strains (Fig. 3E). Consistent with the RNA measurements, flagella exerted an even stronger effect on Tie-$_{2xHA}$ expression (Fig. 3D,F). Taken together, these results show that intact functional flagella are vital for the expression of essential conjugation genes.

## Conjugation genes are activated in a subpopulation in a flagella-dependent manner

To directly test the flagella impact on conjugation gene expression, we monitored conjugation promoter activity. At first, the key conjugation promoter, P$_C$ (Rösch and Graumann, 2015; Ramachandran et al, 2014) (Fig. 3B), was fused to *gfp*, and the reporter was inserted into the host chromosome. Further, given the relatively lengthy intergenic region between ORFs 32 and 33 and the significant decrease observed in RNA levels of downstream genes in the Δ*hag* mutant (Fig. 3B,C), the presence of an additional promoter in this location was investigated. We thus similarly fused this putative promoter to *gfp* and introduced the reporter into the host genome. The activity of the promoters was then evaluated in the absence or presence of pLS20. Fluorescence measurements revealed the existence of an active promoter, although weaker than P$_C$, in the ORFs 32-33 intergenic region, termed P$_{33}$ (Fig. 4A,B). While P$_C$ activity was sharply inhibited in the presence of pLS20, that of P$_{33}$ remained constant (Fig. 4A,B). Surprisingly, the absence or overexpression of flagella appeared to have only a marginal impact, if any, on the overall population fluorescence (Fig. 4A,B).

---

**Figure 3. Flagella affect conjugation gene expression.**

(A) Donor strains: WT (SH337), Δ*hag* (SH443), Δ*CORE* (SH352), and Δ*sigD* (PK143) harboring pLS20$_{cm}$ (-*rapA* OE), and donor strains: WT (SH450), Δ*hag* (SH453), Δ*CORE* (SH451), and Δ*sigD* (SH452), harboring both pLS20$_{cm}$ and amyE::P$_{IPTG}$-*rapA*$_{pLS20}$ (+*rapA* OE), were grown in the presence of IPTG (0.1 mM) and mixed with recipient cells (SH345: *sacA::kan*) in 1:1 ratio. Conjugation mixtures were incubated for 20 min, and serial dilutions were spotted either on LB agar containing chloramphenicol and kanamycin or solely kanamycin, selecting for transconjugants and recipients, respectively. Left panel: Representative images of spotted conjugation mixtures (10$^{-2}$ dilution) over LB agar containing chloramphenicol and kanamycin, selecting for transconjugants. Right panel: Conjugation efficiencies calculated as the number of transconjugants (T)/number of total recipients (T + R), in the presence (blue squares) or absence (red circles) of ectopically expressed *rapA*. Data is presented as scatter dot plot with average values and SEM of three independent biological repeats. Statistical significance was calculated using two-way ANOVA. *P* values of +*rapA* OE vs. -*rapA* OE: (***) = 1 × 10$^{-3}$. (B) Schematic depicting the conjugation operon of pLS20. Regulatory elements controlling pLS20 operon expression, as well as known encoded genes, are highlighted. (C) RNA was isolated from WT (SH337) and Δ*hag* (SH443) donor cells harboring pLS20$_{cm}$ and mRNA levels of the indicated pLS20 genes were determined using qRT-PCR. Shown are relative transcript levels of the indicated genes in Δ*hag* compared to the WT. Data are presented as average values and SEM of at least three independent biological repeats. Statistical significance was calculated using Student's *t* test. *P* values: *rapA* (*) = 3.36 × 10$^{-2}$, *conAn1* (ns not significant) = 9.18 × 10$^{-2}$, *ses* (ns) = 8.32 × 10$^{-2}$, ORF30 (ns) = 2.6 × 10$^{-1}$, ORF31 (*) = 7.24 × 10$^{-3}$, ORF32 (*) = 1.48 × 10$^{-2}$, ORF33 (**) = 2.86 × 10$^{-3}$, *tie* (**) = 4.85 × 10$^{-3}$, *virB11* (***) = 8.7 × 10$^{-4}$, *virD4* (*) = 1.06 × 10$^{-2}$. (D) RNA was isolated from WT (SH337), Δ*hag* (SH443), Δ*CORE* (SH352), Δ*sigD* (PK143), Δ*motA* (SH423), and Δ*flgM* (SH496) donor strains harboring pLS20$_{cm}$, and mRNA level of *tie* was determined using qRT-PCR. Shown are relative transcript levels of *tie* in the indicated mutant strains compared to the WT. Data are presented as average values and SEM of at least three independent biological repeats. Statistical significance was calculated using Student's *t* test. *P* values: Δ*hag* (****) = 7.8 × 10$^{-15}$, Δ*CORE* (****) = 4.45 × 10$^{-14}$, Δ*sigD* (****) = 1.08 × 10$^{-7}$, Δ*motA* (****) = 2.51 × 10$^{-9}$, Δ*flgM* (*) = 1.36 × 10$^{-2}$. (E) Whole-cell lysates were extracted from donor strains: WT (MR13), Δ*hag* (LZ4), Δ*CORE* (LZ5), Δ*sigD* (LZ6), Δ*motA* (LZ7), Δ*flgM* (LZ9), and Δ*ymdB* (LZ8), harboring pLS20$_{cm}$-*conAn1*$_{2xHA}$, and subjected to western blot analysis using anti-HA antibodies (upper panel). Stain-free total protein analysis is presented for comparison (lower panel). ConAn1-$_{2xHA}$ expression levels, normalized to stain-free total protein were calculated as % of WT. Shown is a representative experiment out of three independent biological repeats. (F) Whole-cell lysates were extracted from donor strains: WT (SH461), Δ*hag* (MBS24), Δ*CORE* (MBS26), Δ*sigD* (MBS25), Δ*motA* (SH481), Δ*flgM* (LZ16) and Δ*ymdB* (MBS23), harboring pLS20$_{cm}$-*tie*$_{2xHA}$ and subjected to western blot analysis using anti-HA antibodies (upper panel). Stain-free total protein analysis is presented for comparison (lower panel). Tie-$_{2xHA}$ expression levels, normalized to stain-free total protein were calculated as % of WT. Shown is a representative experiment out of three independent biological repeats. Source data are available online for this figure.

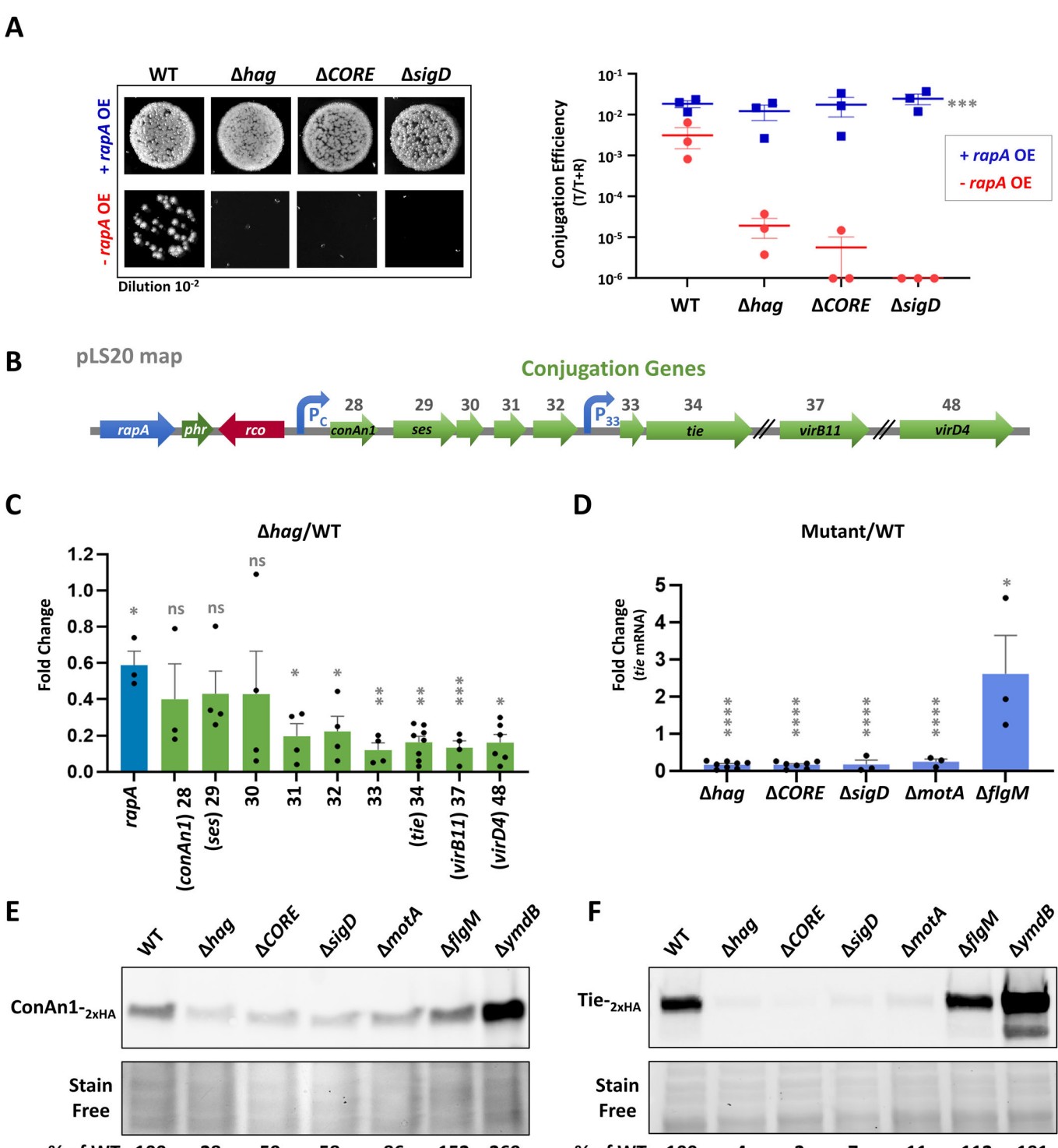

Considering that conjugation is a relatively rare event ($\sim 2.5 \times 10^{-3}$) (Fig. 2E), we tracked the activity of the conjugation promoters at a single-cell resolution using fluorescence microscopy. In the absence of pLS20, both $P_C$ and $P_{33}$ showed uniform levels of GFP expression, irrespective of flagella presence, with $P_C$ exhibiting a significantly higher activity (Fig. 4C,D). In the presence of pLS20, $P_C$ expression was repressed in most bacteria; however, a small subpopulation emerged displaying elevated GFP expression (GFP

"High" cells), which was slightly affected by flagella existence (Fig. 4C–E). Moreover, $P_{33}$ exhibited a sharper flagella-dependent bimodal expression profile, with the "High" population hardly detected in the $\Delta hag$ mutant, yet manifested in the $\Delta ymdB$ hyper-flagellated mutant (Fig. 4C–E).

The strict dependency of $P_{33}$ "High" subpopulation on flagella existence, implies that these cells express the conjugation machinery and could be predictive of forthcoming conjugation events. To investigate

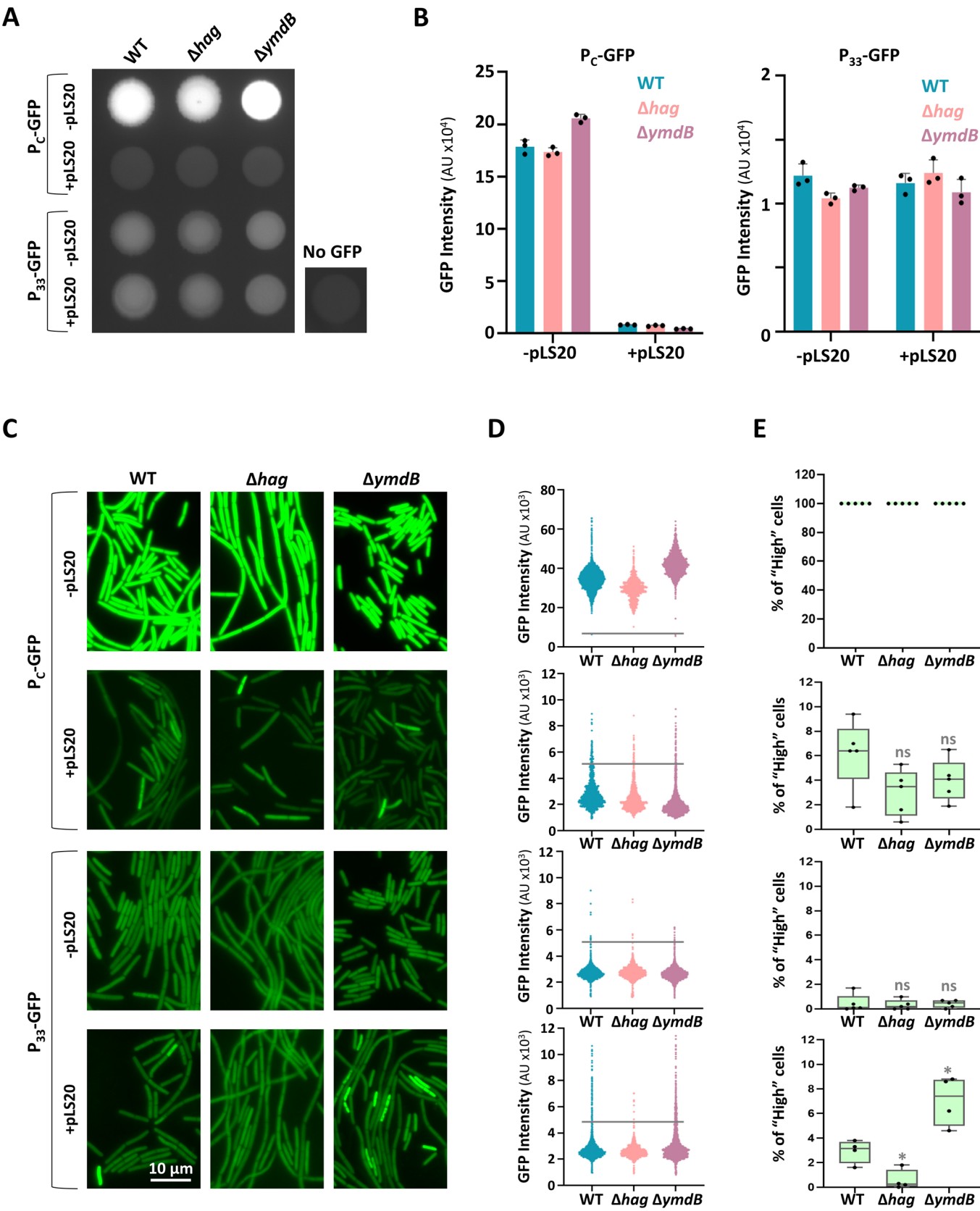

◄ **Figure 4. Conjugation promoters exhibit ON/OFF state.**

(A) Strains harboring $P_C$-*gfp* lacking pLS20$_{cm}$, WT (SH437), Δ*hag* (SH458) and Δ*ymdB* (SH457), or carrying pLS20$_{cm}$, WT (SH444), Δ*hag* (SH448) and Δ*ymdB* (SH447), and strains harboring $P_{33}$-*gfp* lacking pLS20$_{cm}$, WT (SH494), Δ*hag* (SH505) and Δ*ymdB* (SH504), or carrying pLS20$_{cm}$, WT (SH501), Δ*hag* (SH510) and Δ*ymdB* (SH509), were grown in LB to OD$_{600}$ 0.8 and spotted on an LB agar plate. Plates were incubated for 18 h and visualized by a fluorescence imaging system. A WT donor strain (SH337: WT/pLS20$_{cm}$), lacking *gfp* (No GFP), served as a control. Shown is a representative experiment out of three independent biological repeats. (B) Strains in (A) were grown in LB to OD$_{600}$ 0.8, and GFP fluorescence was measured by a fluorescence microplate reader. Shown are normalized GFP intensities in arbitrary units (AU) from $P_C$-GFP (left panel), and $P_{33}$-GFP (right panel). Each bar represents an average value and SEM of three technical replicates. Shown is a representative experiment out of 2 independent biological repeats. (C) Strains in (A) were grown in LB to OD$_{600}$ 0.8, and visualized by fluorescence microscopy. Shown are representative images of GFP fluorescence (green) from $P_C$-GFP (upper panels) and $P_{33}$-GFP (lower panels). Shown is a representative experiment out of three independent biological repeats. (D) Population distribution of cells in (C) based on their GFP expression in arbitrary units (AU). Each dot represents one cell. *n* > 2000 cells for each strain. Horizontal gray lines indicate GFP intensity threshold (5000 AU). Shown is a representative experiment out of three independent biological repeats. (E) Quantification of the percentage of bacterial cells with GFP fluorescence higher than 5000 AU, designated as "High" GFP-expressing cells, in the indicated strains, based on fluorescence microscopy analysis displayed in (D). For each strain, at least three fields were quantified. Data are shown as box plot graphs. The box is determined by the 25th and 75th percentiles, and whiskers are determined by min and max; the line in the box indicates the median. Statistical significance between WT and each mutant was calculated using Student's *t* test. P values: $P_C$-GFP ( + pLS20): Δ*hag* (ns not significant) = 6.44 × 10$^{-2}$, Δ*ymdB* (ns) = 1.67 × 10$^{-1}$; $P_{33}$-GFP (-pLS20): Δ*hag* (ns) = 7.13 × 10$^{-1}$, Δ*ymdB* (ns) = 9.11 × 10$^{-1}$; $P_{33}$-GFP ( + pLS20): Δ*hag* (*) = 9.52 × 10$^{-3}$, Δ*ymdB* (*) = 1.0 × 10$^{-2}$. Source data are available online for this figure.

this prospect, we simultaneously visualized, $P_{33}$-GFP activity in donor cells, conjugation events through the SSB-YFP reporter, and MC formation. Indeed, transconjugants, exhibiting prominent SSB-YFP foci, were predominantly located within MCs containing donor cells exhibiting "High" $P_{33}$-GFP (Fig. 5A), emphasizing that this small subpopulation drives conjugation.

The link between conjugation, flagella, and MC formation raised the possibility that the flagellum-induced *tie*, located downstream to $P_{33}$ (Fig. 3B), previously implicated in mating pair formation (Gago-Córdoba et al, 2021), induces MC assembly. Consistent with earlier studies, a donor carrying a plasmid lacking *tie* (pLS20-Δ*tie*) was impaired in conjugation, a property restored by ectopic *tie* overexpression (Fig. 5B). Biochemical fractionation of donor cells expressing Tie-$_{2xHA}$ indicated the protein to be localized to the cell wall fraction, consistent with its putative role as an adhesin, and according to its expression level in the various flagellar mutants (Figs. 3F and 5C; Appendix Fig. S2D). Moreover, immunofluorescence microscopy exposed the presence of distinct Tie-$_{2xHA}$ foci, decorating the bacterial surface (Figs. 5D and EV1). The presence of Tie-$_{2xHA}$ foci was heterogeneous, with only 3.5% of the cells exhibiting high numbers of foci, in line with $P_C$ and $P_{33}$ expression distribution (Figs. 4E, 5D,E, and EV1). Localization of Tie to the cell surface suggests a possible involvement in MC development. We thus assessed Tie impact on MC formation using quantitative clustering analysis. The clustering efficiency of a *tie* deficient donor (pLS20-Δ*tie*) was slightly reduced, while ectopic expression of *tie* significantly enhanced MC formation in the presence of pLS20 (Fig. 5F; Appendix Fig. S2E), suggesting a modest Tie contribution to this process and the involvement of additional pLS20 factors.

In sum, these findings indicate that flagella induce the expression of conjugation genes in a small subpopulation, demarcated by $P_{33}$ activation, which coincides with MC assembly, and hence conjugation.

## Sensing flagella rotation is required for the expression of conjugation genes

The flagellum harbors a secretory system, which is a key element in the intrinsic circuits hardwired to the process of flagella biogenesis (Appendix Fig. S3A) (Oshiro et al, 2019). Testing if mutants within these circuits expand their scope to regulate conjugation revealed no significant impact on the process (Appendix Fig. S3A,B).

Furthermore, conjugation of Δ*hag* donor could not be rescued in trans by the presence of a hyper-conjugative donor overexpressing RapA (Fig. 6A,B), excluding the involvement of a secreted factor and highlighting that flagella are required strictly in cis.

The finding that the flagellum paralyzed Δ*motA* mutant showed a strong conjugation deficiency that was associated with reduced conjugation gene expression (Figs. 2C–E and 3D–F), raised the possibility that flagellar rotation might be the signal for activation of the conjugation genes. To address this possibility, we grew the bacteria in media containing polyvinylpyrrolidone (PVP), to increase viscosity, and hence the load imposed on the flagella. Indeed, increased viscosity led to a reduction in both conjugation efficiency and Tie-$_{2xHA}$ expression (Fig. 6C,D,J; Appendix Fig. S3C), despite the cells retaining intact flagella (Fig. 6E). Furthermore, a strong reduction in conjugation was evident upon escalating the load on the flagella without affecting viscosity, by growing the bacteria in the presence of anti-Hag antibodies, which crosslink the flagellar filaments (Fig. 6F,J).

To further establish the linkage between flagella rotation and conjugation, we ectopically overexpressed *epsE*, encoding a flagellar clutch, which impedes flagellar rotation (Blair et al, 2008), in donor cells. Remarkably, EpsE induction caused a dramatic reduction in Tie-$_{2xHA}$ expression that was associated with a severe conjugation blockage (Fig. 6G,H,J; Appendix Fig. S3D). As the DegS-DegU two-component system was implicated in relaying flagella rotation to gene expression (Diethmaier et al, 2017; Cairns et al, 2013), the potential involvement of this mechanosensing pathway in conjugation was examined. The conjugation efficiency of a strain lacking the response regulator DegU was similar to that of WT (Fig. 6I,J). Nevertheless, the conjugation defect of the Δ*hag* strain was fully suppressed upon *degU* deletion (Fig. 6I,J), indicating an inhibitory effect of DegU transcriptional factor on conjugation gene expression.

Cumulatively, our results show that upon sensing rotating flagella, a signal is generated to alleviate the inhibitory effect of DegU on conjugation, thereby tying motility with conjugation. The evolution of such plasmid-host molecular cross-talk likely reflects a strategy to activate conjugation when spreading into new habitats is within reach.

## Conjugation–motility coupling fosters spreading into new ecological niches

To demonstrate the prospect that conjugation by flagellated donors is advantageous over non-flagellated ones in promoting invasion into new

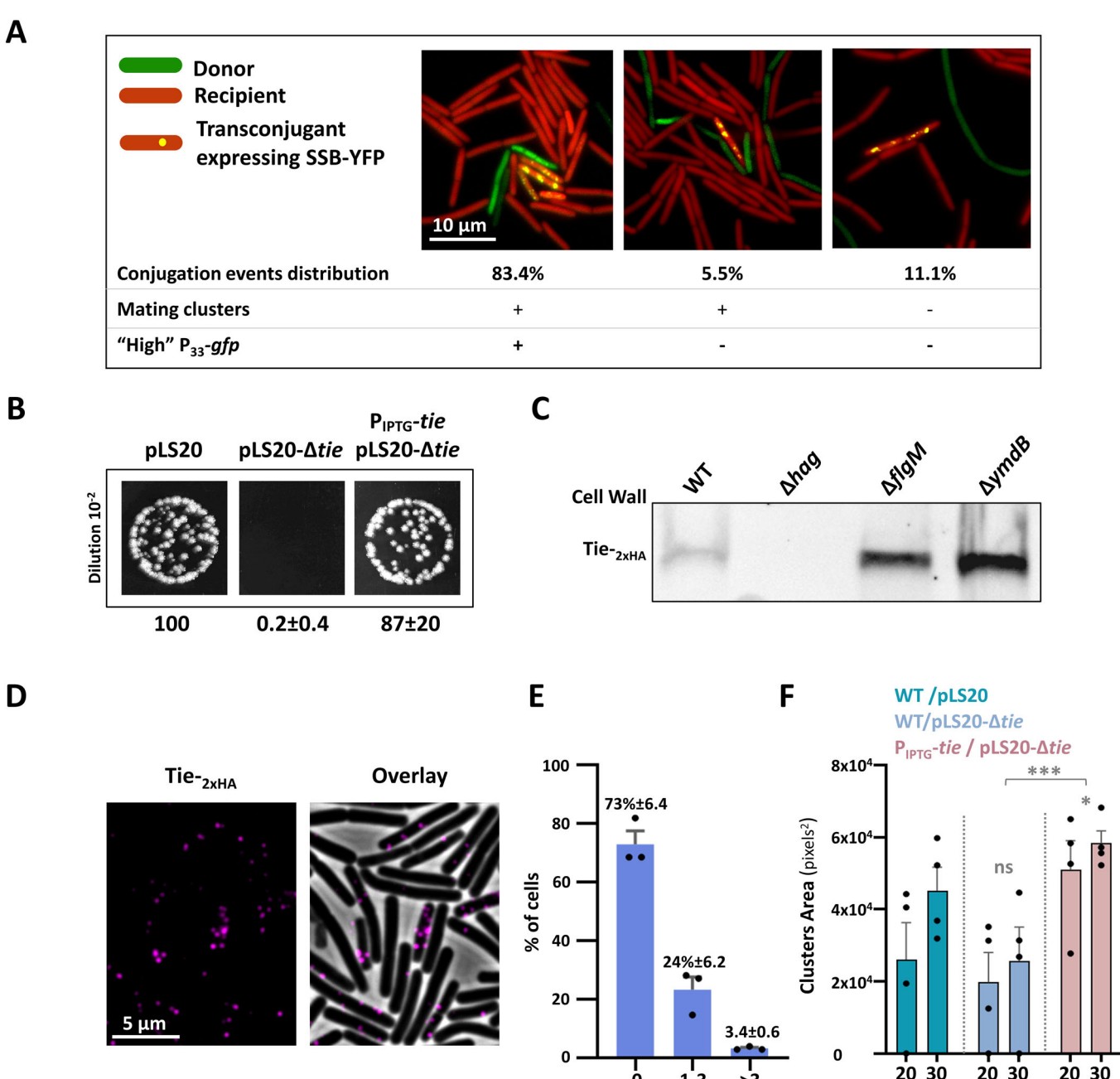

habitats, we exploited the *degU* mutation, which uncouples motility from conjugation gene expression. Accordingly, we compared the spreading efficiency of transconjugants derived from motile (Δ*degU*) and non-motile (Δ*degU* Δ*hag*) donors, having similar conjugation efficiencies (Fig. 6I,J), to a restrictive niche. We simulated an ecological system composed of two niches, one antibiotic-free while the other supplemented with chloramphenicol and spectinomycin, selectively permitting the growth of transconjugants (Fig. 7A). The motile and non-motile donors were spotted over the antibiotic-free zone at equal distances from a motile GFP-expressing recipient, and the expansion of transconjugants into the selective region was monitored (Fig. 7A).

Flagellated donors rapidly encountered the recipients, and transconjugants emerging from the donor-recipient intersection expanded over time into the restrictive zone (Figs. 7B and EV2). In contrast, despite similar conjugation proficiency (Fig. 6I,J), transconjugants derived from the non-motile donor failed to invade the selective zone even after 28 h of incubation (Figs. 7B and EV2). As a control, no expansion into the antibiotic-containing area was seen when monitoring identical strains lacking pLS20 (Fig. 7B). These results insinuate that being motile is beneficial for reaching remotely located recipients and spreading into new niches, highlighting the evolutionary pressure to tie conjugation with flagella rotation.

◀  **Figure 5.  P$_{33}$ "high" state coincides with MC formation and conjugation.**

(A) Donor cells (green) harboring P$_{33}$-*gfp* and pLS20$_{cm}$-*ssb-yfp* (SH579) were mixed with recipient cells (red) (BDR2637: *sacA*::P$_{veg}$-*mCherry*) in 1:1 ratio, placed over semi-solid (0.6%) LB agarose pads, incubated for 80 min, and visualized by fluorescence microscopy. Shown are overlay images of fluorescence from GFP (green), mCherry (red), and SSB-YFP (yellow). Images demonstrating transconjugant cells expressing SSB-YFP were categorized based on MC formation, and the presence of donor cells expressing "High" P$_{33}$-*gfp* (>5000 arbitrary units). The distribution of conjugation events in each category is indicated in the table. Shown is a representative experiment out of two independent biological repeats. (B) Donor strains: WT/pLS20$_{cm}$ (SH337), WT/pLS20$_{cm}$-Δ*tie* (SH483), and a *tie* complementing strain: *amyE*::P$_{IPTG}$-*tie*/pLS20$_{cm}$-Δ*tie* (SH485), were grown in the presence of IPTG (0.5 mM), mixed with recipient cells (SH345: *sacA*::*kan*) in 1:1 ratio, and incubated for 20 min. Shown are images of spotted conjugation mixtures (10$^{-2}$ dilution) over LB agar containing chloramphenicol and kanamycin, selecting for transconjugants. Indicated conjugation efficiencies were calculated as % of WT conjugation efficiency. Presented are average values and SD of at least three independent biological repeats. (C) Donor strains: WT (SH461), Δ*hag* (MBS24), Δ*flgM* (LZ16), and Δ*ymdB* (MBS23), harboring pLS20$_{cm}$-*tie*$_{2xHA}$ were grown to OD$_{600}$ 0.8, cell wall proteins were extracted using LiCl, and samples were subjected to western blot analysis using anti-HA antibodies. Shown is a representative experiment out of three independent biological repeats. (D) Donor cells (SH461: WT/pLS20$_{cm}$-*tie*$_{2xHA}$) were grown to OD$_{600}$ 0.8 and visualized by immunofluorescence microscopy using primary anti-HA antibodies and Alexa647 conjugated secondary antibodies. Cells were not permeabilized before antibody treatment to enable the selective visualization of surface exposed Tie. Shown is an image of fluorescence from Tie$_{-2xHA}$ (magenta), and an overlay image of phase contrast (gray) with fluorescence from Tie$_{-2xHA}$ (magenta). The image of a larger field is presented in Fig. EV1. A representative experiment out of three independent biological repeats. (E) Population distribution of WT donor cells based on the number of Tie$_{-2xHA}$ foci (0, 1–3, >3) displayed on their cell surface, corresponding to (D). $n$ > 1500 cells. Data are presented as average values and SEM of 3 independent biological repeats. (F) Donor strains: WT/pLS20$_{cm}$ (SH337), WT/pLS20$_{cm}$-Δ*tie* (SH483), and a *tie* complementing strain: *amyE*::P$_{IPTG}$-*tie*/pLS20$_{cm}$-Δ*tie* (SH485) were grown in the presence of IPTG (0.5 mM), mixed with recipient cells (BDR2637: *sacA*::P$_{veg}$-*mCherry*) in 1:1 ratio, and the formation of MCs in liquid medium was followed using digital wide-field microscopy. Shown is a quantification of the clustering analysis at the indicated time points after mixing. Representative images for time point 30 min are presented in Appendix Fig. S2E. For each mixture, triplicates were analyzed in parallel, and the total area of clusters larger than 500 pixels$^2$ was calculated. Data are presented as average values and SEM. Statistical significance was calculated using two-way ANOVA. $P$ values: WT/pLS20 vs WT/pLS20-Δ*tie* (ns not significant) = 1.65 × 10$^{-1}$, WT/pLS20 vs P$_{IPTG}$-*tie*/pLS20-Δ*tie* (*) = 2.65 × 10$^{-2}$, WT/pLS20-Δ*tie* vs P$_{IPTG}$-*tie*/pLS20-Δ*tie* (***) = 1.3 × 10$^{-3}$. A representative experiment out of three independent biological repeats. Source data are available online for this figure.

# Discussion

To optimize horizontal distribution, conjugative elements must sense host physiological cues to activate conjugation gene expression under favorable conditions. Here we discovered that pLS20, a prevalent conjugative plasmid found across *Bacillus* species (Val-Calvo et al, 2021a), has evolved to plug-in to the host mechanosensing signal transduction pathway, designed to discern flagella rotation. This adaptation enables the plasmid to coordinate its spread with the motile phase of the bacterial life cycle, which typically coincides with the bacterial necessity to relocate from challenging microenvironments (e.g., Amsler et al, 1993). Furthermore, we demonstrate that this linkage accelerates the dissemination of antibiotic resistance through conjugation. Upon detecting propelled flagella, pLS20 conjugative genes, including the T4SS machinery, are induced in a subset of cells, becoming primed to exploit potential conjugation opportunities. As conjugation demands a fastened donor-recipient pairing to allow DNA delivery, pLS20 orchestrates the clustering of the donor with recipient bacteria, aided by the plasmid-encoded Tie protein. These MCs facilitate intimate proximity between donor and recipient bacteria, providing a hub for mating pair formation (Fig. 7C). Interestingly, foreign species could be efficiently recruited to *B. subtilis* donor-derived MCs, alluding to the potential of these dynamic cellular assemblages to drive genetic exchange among species within natural niches (Fig. 7C).

Most well-studied conjugation systems favor solid surfaces, offering stability and high cell density, crucial for mating pair establishment and plasmid delivery (Grohmann et al, 2003). In fact, biofilm cellular assemblages emerge as significant hotspots for horizontal gene transfer among bacteria residing in natural habitats (e.g., Madsen et al, 2012; Lécuyer et al, 2018), a phenomenon that can even be promoted by artificial bacterial adhesion (Robledo et al, 2022). Consistently, some conjugative elements were found to actively temper host motility, possibly facilitating the establishment of cell-to-cell connection. A prominent example is the enterobacteria conjugative plasmid R27,

encoding factors that downregulate flagella synthesis (Luque et al, 2019). Surprisingly, pLS20 has evolved to synchronize activation of conjugation with functional flagella, although *B. subtilis* cells are known to form robust biofilms (e.g., Kearns and Losick, 2005; Vlamakis et al, 2013; Arnaouteli et al, 2021). This feature provides the plasmid with the benefit of spreading into remote destinations, seizing opportunities to invade new hosts of the same or diverse species. Although we could not detect pLS20 exchange between *Bacilli* species, its ability to mobilize other plasmids as well as its prevalence among *Bacilli* (Koehler and Thorne, 1987; Val-Calvo et al, 2021a), supports the occurrence of such events in nature. The strategy of inducing the formation of temporal MCs can be advantageous over biofilms, which confine donor bacteria to relatively predetermined, fixed niches.

The capacity of mobile genetic elements to be tuned with the host physiological state is a prevalent trait that appears to maximize their fitness. Numerous prophages, such as lambda, trigger the lysogeny-to-lysis transition in response to host DNA damage, a synchronization facilitated through host SOS regulatory factors (Brady et al, 2021; Wolfe et al, 2014). Likewise, the excision of the *B. subtilis* mobile genetic element, ICE*Bs1*, was shown to be induced by the global DNA-damage SOS response (Auchtung et al, 2005), whereas the *tra* operon of the F-family conjugative plasmid is governed by the host aerobic respiration control factor ArcA, hence directly linking conjugation with oxygen levels (Lu et al, 2019). Importantly, conjugative elements can also interfere with host physiology by modifying global gene expression, and may even delay entry into biofilm or sporulation developmental states (Jones et al, 2021; Miyazaki et al, 2018; Rösch et al, 2014). Similarly, transcriptome analysis of *B. subtilis* cells harboring pLS20, revealed a substantial impact on host gene expression, including genes involved in metabolic pathways, cell envelope, motility, signal transduction, and regulatory pathways (Rösch et al, 2014). Thus, mobile genetic elements can sense and intervene with host physiology.

An important facet of conjugative plasmids is their maintenance in a default 'OFF' state, presumably to avoid the fitness cost that would be associated with their constitutive expression (Koraimann

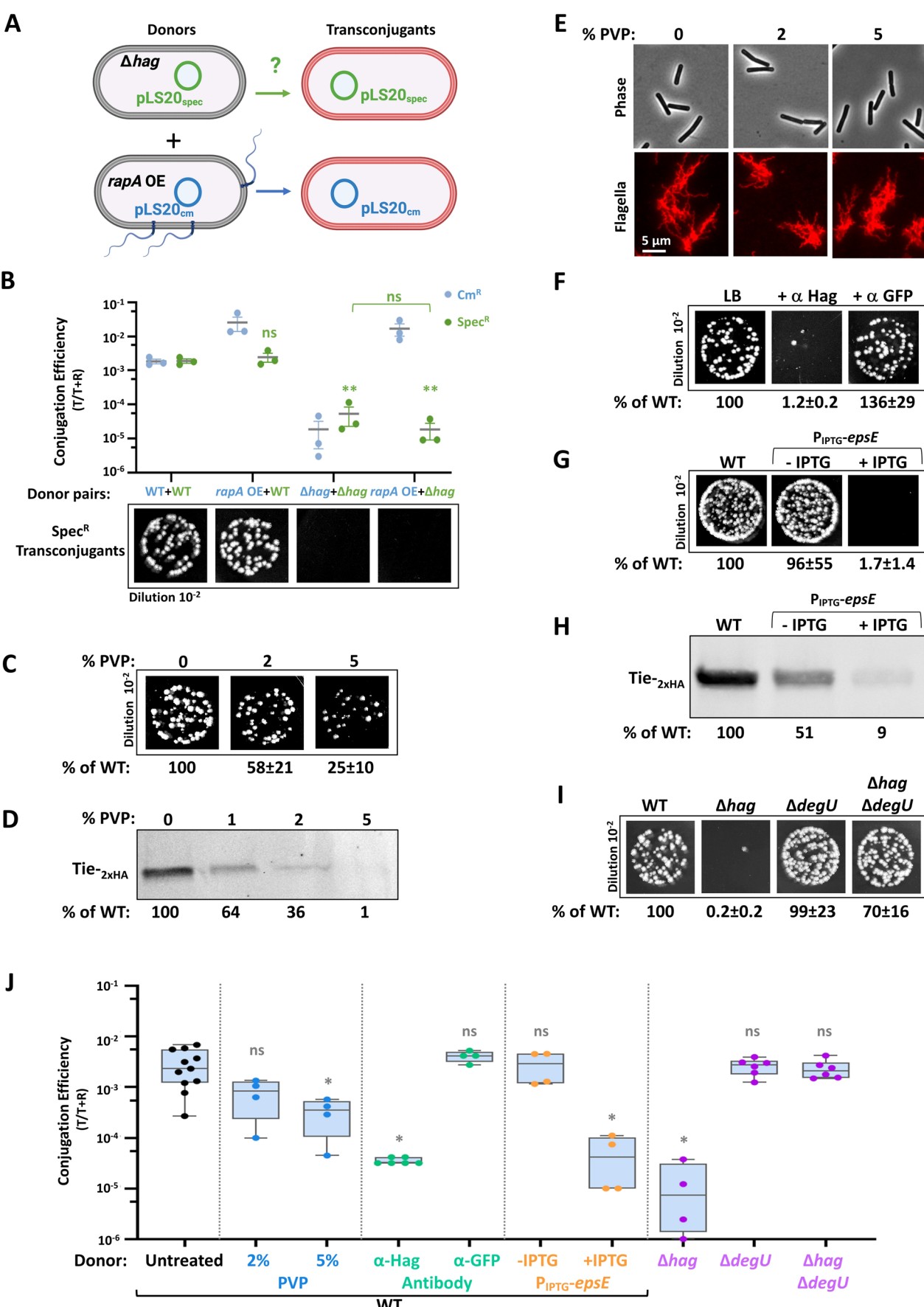

**Figure 6.  Flagella rotation dictates pLS20 conjugation.**

(A) Schematics describing the experimental design to test the ability of a donor strain harboring pLS20$_{cm}$ and overexpressing *rapA* (*rapA* OE) to complement in trans the conjugation defect of a Δ*hag* donor strain harboring pLS20$_{spec}$. Conjugation was assayed by monitoring both spectinomycin-resistant transconjugants and chloramphenicol-resistant transconjugants. Created in BioRender. Ben-Yehuda, S. (2024) BioRender.com/g81h655. (B) Corresponds to the experimental design described in (A). Pairs of donor strains: WT (SH337: WT/pLS20$_{Cm}$) with WT (SH342: WT/pLS20$_{spec}$), *rapA* OE (SH450: *amyE*::P$_{IPTG}$-*rapA*/pLS20$_{Cm}$) with WT (SH342: WT/pLS20$_{spec}$), Δ*hag* (SH443: Δ*hag*/pLS20$_{Cm}$) with Δ*hag* (MBS17: Δ*hag*/pLS20$_{spec}$), and *rapA* OE (SH450: *amyE*::P$_{IPTG}$-*rapA*/pLS20$_{Cm}$) with Δ*hag* (MBS17: Δ*hag*/pLS20$_{spec}$), were grown in the presence of IPTG (0.1 mM), mixed with recipient cells (SH345: *sacA*::*kan*) in 1:1:2 (donor: donor: recipient) ratio, and incubated for 20 min. Serial dilutions were then spotted on LB agar containing chloramphenicol and kanamycin, spectinomycin and kanamycin, or solely kanamycin, to specifically select for pLS20$_{cm}$ or pLS20$_{spec}$ derived transconjugants and recipients, respectively. Upper panel: Conjugation efficiencies calculated as the number of pLS20$_{cm}$ (blue) or pLS20$_{spec}$ (green)-derived transconjugants (T)/number of total recipients (T + R). Data shown as scatter dot plot with average values and SEM of three independent biological repeats. Statistical significance between conjugation efficiencies of pLS20$_{spec}$ derived from the indicated donor pairs was calculated using Student's *t* test. *P* values: *rapA* OE + WT (ns not significant) $= 5.22 \times 10^{-1}$, Δ*hag* + Δ*hag* (**) $= 2.41 \times 10^{-3}$, *rapA* OE + Δ*hag* (**) $= 2.2 \times 10^{-3}$; Δ*hag* + Δ*hag* vs *rapA* OE + Δ*hag* (ns)$= 3.35 \times 10^{-1}$. Lower panel: Representative images of spotted conjugation mixtures ($10^{-2}$ dilution) over LB agar containing spectinomycin and kanamycin, selecting for pLS20$_{spec}$ transconjugants. (C) Donor (SH337: WT/pLS20$_{cm}$) and recipient (SH345: *sacA*::*kan*) cells were grown in LB medium supplemented with the indicated concentrations of polyvinylpyrrolidone 360 (PVP), mixed in 1:1 ratio, incubated for 20 min, and serial dilutions were spotted either on LB agar containing chloramphenicol and kanamycin or solely kanamycin, for transconjugants and recipients, respectively. Shown are images of spotted conjugation mixtures ($10^{-2}$ dilution) over LB agar containing chloramphenicol and kanamycin, selecting for transconjugants. Indicated conjugation efficiencies were calculated as % of WT conjugation efficiency. Presented are average values and SD of at least three independent biological repeats. (D) Whole-cell lysates were extracted from WT donor cells (SH461: WT/pLS20$_{cm}$-*tie*-$_{2xHA}$) grown in LB medium supplemented with indicated concentrations of PVP and subjected to western blot analysis using anti-HA antibodies. Corresponding stain-free total protein analysis is presented for comparison in Appendix Fig. S3C. Tie-$_{2xHA}$ expression levels, normalized to stain-free total protein, were calculated as % of 0 PVP. Shown is a representative experiment out of 3 independent biological repeats. (E) Cells harboring modified flagellin *hag*$^{T209C}$ (DS1895: *amyE*::P$_{hag}$-*hag*$^{T209C}$) were grown in LB medium supplemented with indicated concentrations of PVP, flagellin was stained with Alexa Fluor 594 C$_5$ maleimide, and cells were visualized by fluorescence microscopy. Shown are images of phase contrast (gray) (upper panels) and corresponding stained flagella (red) (lower panels). (F) Donor cells (SH337: WT/pLS20$_{cm}$) were grown in LB medium supplemented with anti-Hag or anti-GFP antibodies as indicated, mixed with recipient cells (SH345: *sacA*::*kan*) in 1:1 ratio, and incubated for 20 min. Shown are images of spotted conjugation mixtures ($10^{-2}$ dilution) over LB agar containing chloramphenicol and kanamycin, selecting for transconjugants. Indicated conjugation efficiencies were calculated as % of WT conjugation efficiency. Presented are average values and SD of at least 3 independent biological repeats. (G) Donor strains: WT (SH337: WT/pLS20$_{cm}$) and P$_{IPTG}$-*epsE* (LZ53: *amyE*::P$_{IPTG}$-*epsE*/pLS20$_{cm}$) were grown in the absence or presence of IPTG (0.5 mM) as indicated, mixed with recipient cells (SH345: *sacA*::*kan*) in 1:1 ratio, and incubated for 20 min. Shown are images of spotted conjugation mixtures ($10^{-2}$ dilution) over LB agar containing chloramphenicol and kanamycin, selecting for transconjugants. Indicated conjugation efficiencies were calculated as % of WT conjugation efficiency. Presented are average values and SD of at least three independent biological repeats. (H) Whole-cell lysates were extracted from donor strains: WT (SH461: WT/ pLS20$_{cm}$-*tie*-$_{2xHA}$) and P$_{IPTG}$-*epsE* (SH582: *amyE*::P$_{IPTG}$-*epsE*/pLS20$_{cm}$-*tie*-$_{2xHA}$) grown in the absence or presence of IPTG (1 mM) as indicated, and subjected to western blot analysis using anti-HA antibodies. Stain-free total protein analysis is presented for comparison in Appendix Fig. S3D. Tie-$_{2xHA}$ expression levels, normalized to stain-free total protein, were calculated as % of WT. Shown is a representative experiment out of 3 independent biological repeats. (I) Donor strains: WT (SH337), Δ*hag* (SH443), Δ*degU* (SH592), and Δ*hag* Δ*degU* (SH593) harboring pLS20$_{cm}$, were mixed with recipient cells (SH360: *sacA*::*spec*) in 1:1 ratio, incubated for 20 min, and serial dilutions were spotted either on LB agar containing chloramphenicol and spectinomycin or solely spectinomycin, selecting for transconjugants and recipients, respectively. Shown are images of spotted conjugation mixtures ($10^{-2}$ dilution) over LB agar containing chloramphenicol and spectinomycin, selecting for transconjugants. Indicated conjugation efficiencies were calculated as % of WT conjugation efficiency. Presented as average values and SD of at least three independent biological repeats. (J) Conjugation efficiencies derived from the experiments presented in Fig. 6C,F,G,I, calculated as the number of transconjugants (T)/number of total recipients (T + R). For each strain, at least four independent biological repeats were conducted. Data are shown as box plot graphs. The box is determined by the 25th and 75th percentiles, and whiskers are determined by min and max; the line in the box indicates the median. Statistical significance between untreated WT and each treatment/mutant was calculated using Student's *t* test. *P* values: 2% PVP (ns not significant) $= 7.77 \times 10^{-2}$, 5% PVP (*) $= 3.69 \times 10^{-2}$, α-Hag (*) $= 1.15 \times 10^{-2}$, α-GFP (ns) $= 3.73 \times 10^{-1}$, P$_{IPTG}$-*epsE* (-IPTG) (ns) $= 9.3 \times 10^{-1}$, P$_{IPTG}$-*epsE* (+IPTG) (*) $= 2.32 \times 10^{-2}$, Δ*hag* (*) $= 2.18 \times 10^{-2}$, Δ*degU* (ns) $= 7.09 \times 10^{-1}$, and Δ*hag* Δ*degU* (ns) $= 5.18 \times 10^{-1}$. Source data are available online for this figure.

and Wagner, 2014; Kohler et al, 2019). Regulation of pLS20 has been primarily attributed to the modulation of the P$_C$ promoter activity (Rösch and Graumann, 2015; Ramachandran et al, 2014). However, the discovery of additional promoters, including P$_{33}$ in this study, along with transcription terminators within the conjugation operon of pLS20 (Miguel-Arribas et al, 2023, 2021; Gago-Córdoba et al, 2019), highlight the presence of additional layers of regulation, facilitating a switch from conjugation 'OFF' to 'ON' state. Fascinatingly, pLS20 senses the host flagellar rotation that serves as a mechanotransmitter to activate P$_C$ and P$_{33}$ promoters in a subset of cells, via the DegS-DegU system. Interestingly, *B. subtilis* flagella has been implicated as a mechanotransmitter, dictating developmental processes such as competence and biofilm formation (Diethmaier et al, 2017; Cairns et al, 2013; Hölscher et al, 2018). Bacterial mechanosensing appears as a widespread strategy for integrating environmental cues to coordinate complex physiological responses (e.g., Lele et al, 2013; Persat, 2017). Mechanosensing properties have also been attributed to type IV pili in *Pseudomonas aeruginosa* in detecting solid surfaces and host cells to activate virulence pathways (Persat et al, 2015a, 2015b; Siryaporn et al, 2014; Persat, 2017), whereas in *Caulobacter*

*crescentus*, surface sensing can be mediated both via the tight adherence pilus and flagella (Hershey et al, 2021; Snyder et al, 2020).

Flagella have also been implicated in serving as an adhesin in several bacteria, including pathogenic *E. coli*, *P. aeruginosa*, *Salmonella* Typhimurium, *Clostridium difficile*, mediating contact with host cells or even with abiotic surfaces (e.g., Tasteyre et al, 2001; Friedlander et al, 2015; Roy et al, 2009; Lillehoj et al, 2002; Horstmann et al, 2020; Haiko and Westerlund-Wikström, 2013; Serra et al, 2013). Further, flagella of the marine bacterium *P. marina* have been demonstrated to promote intraspecies aggregation in artificial sea water (Sjoblad et al, 1985). It is conceivable that flagella might mediate an initial contact during pLS20 conjugation, enhancing donor-recipient proximity, aiding Tie to securely fasten bacterial cells to form stable MCs and hence mating pair formation. However, according to our data, flagella contribution to MC is restricted to donor bacteria, suggesting that in this case flagella by itself cannot simply act as an aggregation facilitator.

Our findings unravel the riddle of efficient conjugation of pLS20 in liquid medium, thereby shedding light on novel mechanisms

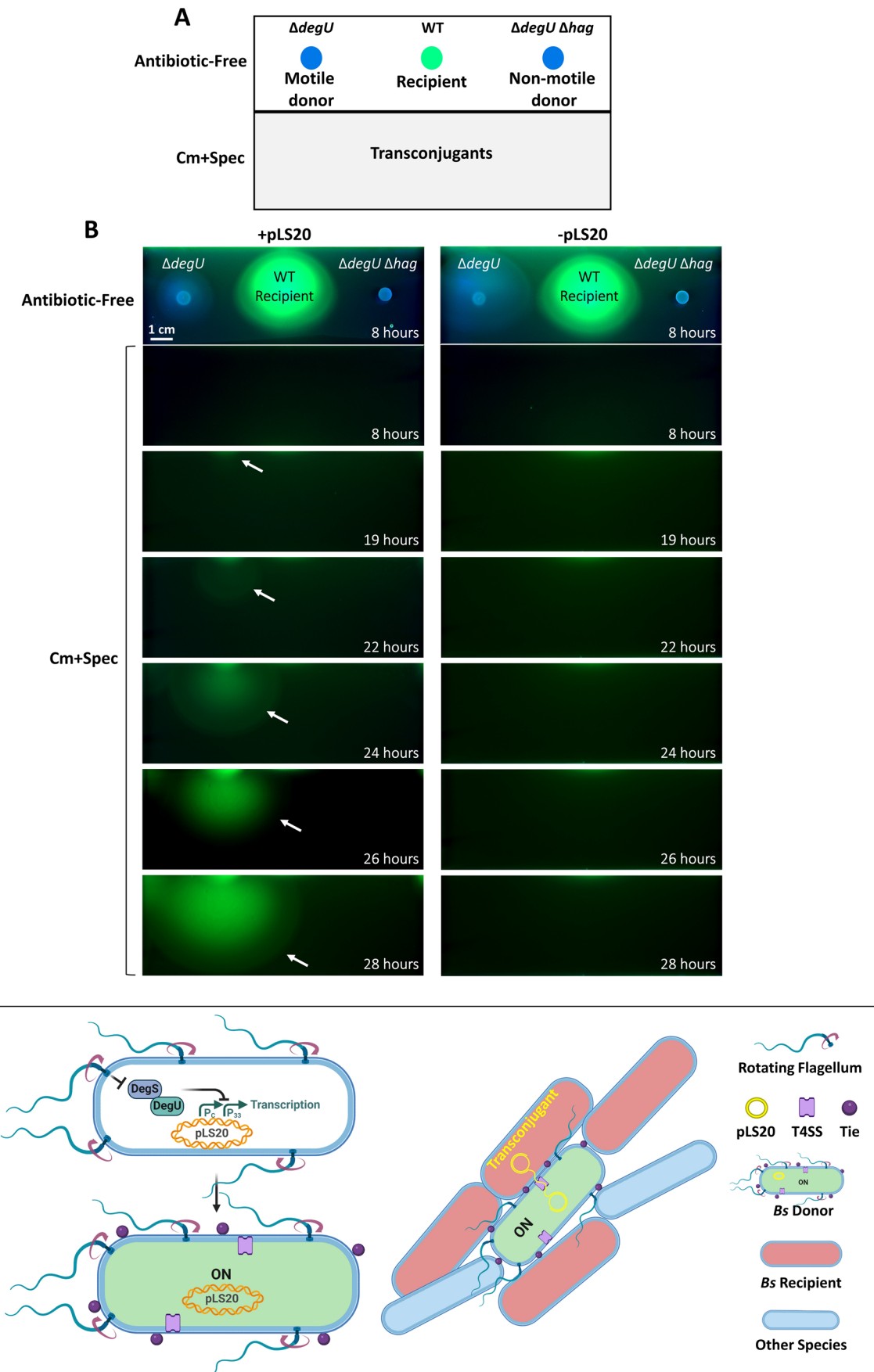

◄ **Figure 7. The flagella-conjugation link allows spreading into new ecological niches.**

(A) A diagram outlining the experimental design to demonstrate the impact of flagella on pLS20$_{cm}$ spread. A split-plate soft agar assay was devised, wherein the top half of a rectangular plate contained antibiotic-free LB, while the bottom half contained LB supplemented with chloramphenicol and spectinomycin (Cm+Spec), selecting for transconjugants. The entire plate was layered with soft agar, and motile (Δ*degU*) and non-motile (Δ*hag* Δ*degU*) donors (blue circles) were spotted at equal distances from a WT motile GFP-expressing recipient (green circle) on the antibiotic-free zone. The soft agar allows bacterial growth, swimming motility, and conjugation. The emergence of motile GFP-expressing transconjugants resistant to chloramphenicol and spectinomycin on the antibiotic-selective region could then be monitored over time. (B) Corresponds to the experimental design in (A). Bacteria were grown to mid-logarithmic phase and spotted on the antibiotic-free region of the plate. Left panel: plates were spotted with Δ*degU* (SH592: Δ*degU*/pLS20$_{cm}$), Δ*hag* Δ*degU* (SH593: Δ*hag* Δ*degU*/pLS20$_{cm}$), and WT (AR16: P$_{rrnE}$-*gfp*) strains. Right panel: plates were spotted with Δ*degU* (SH590: Δ*degU*), Δ*hag* Δ*degU* (SH609: Δ*hag* Δ*degU*), and WT (AR16: P$_{rrnE}$-*gfp*) strains. Presented are overlay images of colorimetric channel (blue) with fluorescence from GFP (green), captured at the indicated time points post incubation using ChemiDoc MP imaging system. Upper panels: images from the antibiotic-free region after 8 h of incubation. Lower panels: a series of images from the antibiotic-containing region (Cm+Spec), captured at the indicated time points post incubation. White arrows point at the emergence and spread of transconjugants. Images of the entire plates (left panels) are presented in Fig. EV2. Shown is a representative experiment out of three independent biological repeats. (C) A model depicting donor flagella-driven activation of conjugation gene expression and MC formation in liquid milieu. Left Panel: schematic illustrating a motile *B. subtilis* (*Bs*) donor cell, in which flagellar rotation leads to alleviation of the inhibitory effect of DegS-DegU two-component system on pLS20. Consequently, transcription of crucial conjugation genes, including the T4SS machinery and Tie, regulated by P$_C$ and P$_{33}$ are activated, thereby switching to a conjugation "ON" state. Right Panel: Assembly of MC, composed of *B. subtilis* and other species, induced by a conjugation primed "ON" cell, providing a stable platform for mating pair formation and plasmid DNA transfer. Created in BioRender. Ben-Yehuda, S (2024) BioRender.com/g81h655. Source data are available online for this figure.

driving horizontal gene transfer in microbial communities residing in fluid environments. The implications of this study could inform strategies for controlling the spread of antibiotic resistance genes through conjugation in Gram-positive bacteria.

# Methods

## Reagents and tools table

## Bacterial strains and plasmids

All *B. subtilis* strains used in this study are derivatives of wild-type PY79 strain (Youngman et al, 1984). *B. cereus* (OS4) *and B. megaterium* (OS2) (Stempler et al, 2017) are wild-type soil isolates (laboratory stock). All bacterial strains used in this study are described in Table EV1 (i.e. Dubey and Ben-Yehuda 2011; Koo

| Reagent/resource | Reference or source | Identifier or catalog number |
|---|---|---|
| **Experimental models** | | |
| Bacterial strains | This study | Table EV1 |
| **Recombinant DNA** | | |
| pDR111 (*amyE*::P$_{hyper-spank}$-*spec*) | Kindly provided by Prof. David Rudner (Harvard U) | N/A |
| pDR244 (P$_{PA}$-*cre-ori(ts)-spec*) | Kindly provided by Prof. David Rudner (Harvard U) | N/A |
| pWX466 (loxP *spec*) | Kindly provided by Prof. David Rudner (Harvard U) | N/A |
| pWX467 (loxP *erm*) | Kindly provided by Prof. David Rudner (Harvard U) | N/A |
| pWX470 (loxP *kan*) | Kindly provided by Prof. David Rudner (Harvard U) | N/A |
| pLZ3 (*amyE*::P$_{hyper-spank}$-*epsE-spec*) | This study | Constructed by amplifying *epsE* from PY79 genomic DNA using primers 7478-7479. The PCR-amplified DNA was cloned into pDR111 digested with *Hind*III and *Sph*I using Gibson assembly master mix. |
| pSH46 (*amyE*::P$_{hyper-spank}$-*gfp*$_{-2xHA-SpyTag003}$-*spec*) | Laboratory stock | N/A |
| pSH55 (*amyE*::P$_{hyper-spank}$-*rapA*$_{pLS20}$-*spec*) | This study | Constructed by amplifying *rapA*$_{pLS20}$ from MBS7 DNA using primers 5419-5420. The PCR-amplified DNA was cloned into pDR111 digested with *Hind*III and *Sph*I using Gibson assembly master mix. |
| pSH58 (*amyE*::P$_{hyper-spank}$-*tie*$_{pLS20}$-*spec*) | This study | Constructed by amplifying *tie*$_{pLS20}$ from MBS7 DNA using primers 5642-5643. The PCR-amplified DNA was cloned into pDR111 digested with *Hind*III and *Sph*I using Gibson assembly master mix. |
| **Antibodies** | | |
| HA Tag Polyclonal Antibody (Rabbit) | Thermo Fisher Scientific | Cat#: 71-5500 |
| GFP Polyclonal Antibody (Rabbit) | Thermo Fisher Scientific | Cat#: PA1-980A |
| GFP Monoclonal Antibody (Mouse) | Thermo Fisher Scientific | Cat#: MA5-15256 |

| Reagent/resource | Reference or source | Identifier or catalog number |
|---|---|---|
| Anti-Hag | Gift from Daniel B. Kearns | |
| Donkey Anti-Rabbit IgG H&L (Alexa Fluor® 647) | Abcam | Cat#: ab150075 |
| Goat Anti-Rabbit IgG StarBright™ Blue 700 | Bio-Rad | Cat#: 12004161 |
| Goat Anti-Mouse IgG StarBright™ Blue 520 | Bio-Rad | Cat#: 12005867 |
| **Oligonucleotides and other sequence-based reagents** | | |
| PCR primers | This study | Table EV2 |
| **Chemicals, enzymes, and other reagents** | | |
| Chloramphenicol | Sigma-Aldrich | Cat#: C0378 |
| Kanamycin | US Biological | Cat#: K0010 |
| Lincomycin | Sigma-Aldrich | Cat#: 62143-5G |
| Erythromycin | Sigma-Aldrich | Cat#: E0774 |
| Spectinomycin | Sigma-Aldrich | Cat#: S4014-5G |
| Ampicillin | Sigma-Aldrich | Cat#: A9518-25G |
| Tetracycline | Sigma-Aldrich | Cat#: T3258 |
| Isopropyl-β-D-thiogalactopyranoside (IPTG) | Sigma-Aldrich | Cat#: I1284-5ML |
| Phosphate Buffered Saline (PBS, X10) | Biological Industries | Cat#: 02-020-1A |
| Tris-Buffered Saline (TBS, X10) | BioPrep | Cat#: TBS-VE4-1L |
| Halt™ Protease Inhibitor Cocktail (X100) | Thermo Fisher Scientific | Cat#: 78438 |
| 4x Laemmli sample buffer | Bio-Rad | Cat#: 1610747 |
| BD Difco™ LB broth, Lennox | BD Biosciences | Cat#: 240230 |
| Paraformaldehyde 16% | Electron Microscopy Sciences | Cat#: 15710 |
| Gibson Assembly® Master Mix | NEW ENGLAND BioLabs | Cat#: E2611L |
| Polyvinylpyrrolidone 360 (PVP) | Sigma-Aldrich | Cat#: PVP360 |
| Verifi™ Hot Start Mix | PCR BIOSYSTEMS | Cat#: PB10.46-02 |
| UltraPure™ Agarose | Thermo Fisher Scientific | Cat#: 16500500 |
| TURBO™ DNA free kit | Thermo Fisher Scientific | Cat#: AM1907 |
| iTaq™ Universal SYBR® Green Supermix | Bio-Rad | Cat#: 1725124 |
| DNAseI | Roche | Cat#: 10104159001 |
| Alexa Fluor™ 594 C$_5$ Maleimide | Thermo Fisher Scientific | Cat#: A10256 |
| Trichloroacetic acid (TCA) | Sigma-Aldrich | Cat#: T0699 |
| Lysozyme | GOLDBIO | Cat#: L-040-5 |
| Lithium Chloride | J.T.Baker | Cat#: JTB-2370-01 |
| Difco™ Skim Milk | BD Biosciences | Cat#: 232100 |
| **Software** | | |
| Bio-Rad CFX Maestro software | https://www.bio-rad.com/en-il/product/cfx-maestro-software-for-cfx-real-time-pcr-instruments?ID=OKZP7E15 | |
| GraphPad Prism | https://www.graphpad.com/ | |
| FIJI | https://imagej.net/ | |
| NIS-Elements AR | Nikon https://www.microscope.healthcare.nikon.com/en_EU/products/software/nis-elements | |

                                                    

| Reagent/resource | Reference or source | Identifier or catalog number |
|---|---|---|
| MetaMorph Microscopy Automation and Image Analysis Software | Molecular devices https://www.moleculardevices.com/products/cellular-imaging-systems/high-content-analysis/metamorph-microscopy | |
| BioTek Gen5 | https://www.agilent.com/en/product/microplate-instrumentation/microplate-instrumentation-control-analysis-software/imager-reader-control-analysis-software/biotek-gen5-software-for-detection-1623227 | |
| SparkControl | https://lifesciences.tecan.com/multimode-plate-reader?p=tab--3 | |
| Primer3 | https://primer3.ut.ee/ | |
| Snapgene | http://www.snapgene.com/ | |
| BioRender | https://www.biorender.com/ | |
| **Other** | | |
| 4–20% Mini-PROTEAN® TGX Stain-Free™ Protein Gels | Bio-Rad | Cat#: 4568094 |
| Trans-Blot Turbo Transfer Pack, 0.2 µm nitrocellulose membrane | Bio-Rad | Cat#: 1704158 |
| Direct-zol™ RNA MiniPrep Plus Kit | ZYMO RESEARCH | Cat#: R2072 |
| iScript™ cDNA Synthesis Kit | Bio-Rad | Cat#:170891 |

et al, 2017; Rosenberg et al, 2012; Tzipilevich et al, 2017). All plasmids used in this study are described in the Reagents and Tools Table. All primers used in this study are listed in Table EV2. Plasmid constructions were performed using standard molecular biology methods in *E. coli* DH5α.

## General growth conditions

All general methods for *B. subtilis* were carried out as described previously (Harwood and Cutting, 1990). *B. subtilis* cultures were inoculated from an overnight culture and growth was carried out to the desired $OD_{600}$ at 37 °C in LB medium (BD Biosciences). For strains harboring genes under inducible promoters, 0.1–1 mM Isopropyl-β-D-thiogalactopyranoside (IPTG, Sigma-Aldrich) was added to the medium, as indicated. Antibiotics were used at the following concentrations: chloramphenicol 5 µg/ml (Sigma-Aldrich), kanamycin 10 µg/ml (US Biological), erythromycin 1 µg/ml (Sigma-Aldrich), lincomycin 25 µg/ml (Sigma-Aldrich), spectinomycin 100 µg/ml (Sigma-Aldrich), tetracycline 10 µg/ml (Sigma-Aldrich). For *E. coli* strains, ampicillin 100 µg/ml (Sigma-Aldrich) was used for selection.

## Conjugation assay by CFU

Conjugation assay in liquid medium was based on a previously described protocol (Singh et al, 2013). *B. subtilis* donor cells, harboring pLS20 or its derivatives, and recipient cells, lacking a plasmid, were grown to $OD_{600}$ 0.8 at 37 °C in LB medium, mixed in 1:1 ratio and incubated for 20 min at 37 °C under static conditions to permit conjugation. Serial dilutions were spotted (100 µl) on LB agar containing appropriate antibiotics to specifically select for transconjugants, donors, and recipients. CFUs were counted from dilutions showing well-separated colonies. Conjugation efficiencies were calculated as number of Transconjugant CFUs (T)/number of total recipient CFUs, including transconjugants (T + R).

### Conjugation in the presence of PVP
*B. subtilis* donor cells, harboring pLS20, and recipient cells, lacking a plasmid, were grown to $OD_{600}$ 0.8 at 37 °C in LB medium supplemented with different Polyvinylpyrrolidone 360 (PVP, Sigma-Aldrich) concentrations to increase the viscosity of the medium. Conjugation assay was conducted in the presence of PVP as described above.

### Conjugation in the presence of antibodies
*B. subtilis* donor cells, harboring pLS20 were grown for 2 h at 37 °C in LB medium supplemented with anti-Hag (a gift from Daniel B. Kearns) or anti-GFP antibodies (Thermo Fisher Scientific) at a final concentration of 1:100. In parallel, WT recipient cells, lacking a plasmid, were grown to $OD_{600}$ 0.8 at 37 °C in LB medium. Conjugation assay was then conducted as described above.

## Split-plate soft agar conjugation assay

For conjugation on soft agar, freshly prepared LB containing 0.3% agar was layered over an LB-rectangular plate containing 1.5% agar that was split into two halves. The top half contained antibiotic-free LB, permitting bacterial growth, swimming motility, and conjugation. The bottom part contained LB supplemented with chloramphenicol and spectinomycin antibiotics for the selection of transconjugants. The combined plates were composed by excising the LB bottom half, and replacing it with LB supplemented with chloramphenicol and spectinomycin. Strains harboring or lacking pLS20, and recipient cells constitutively expressing GFP, were grown to mid-logarithmic phase in LB and concentrated to $OD_{600}$ 5.0. Cell suspension (3 µl) was spotted over the antibiotic-free region. Plates were incubated at 37 °C and imaged over time using ChemiDoc MP imaging system (Bio-Rad).

## Motility assay

Motility assay was based on a previously described protocol (Kearns and Losick, 2003). Cells were grown to mid-logarithmic phase in LB

and concentrated to $OD_{600}$ 5.0. Cell suspension (5 μl) was spotted on freshly prepared LB plates containing 0.3% agar, incubated at 37 °C for 7 h, and imaged using Epson Perfection V800 Photo scanner.

## Protein extraction and western blot analysis

### Protein extraction

For the preparation of whole-cell lysates, *B. subtilis* cells were grown to $OD_{600}$ 0.8 at 37 °C in LB medium. Cells were harvested and suspended in lysis buffer containing 10 mM Tris-HCl pH 8.0, 10 mM $MgCl_2$, 0.5 mg/ml Lysozyme, 5 μg/ml DNaseI, and 1x Halt™ protease inhibitor cocktail, and incubated at 37 °C for 10 min. Following cell lysis, samples were incubated at 95 °C for 10 min with 1× Laemmli sample buffer.

Cell wall protein extraction was carried out as described previously (Antelmann et al, 2002). Briefly, *B. subtilis* cells were grown to $OD_{600}$ 0.8 at 37 °C in LB medium. Cells were harvested and washed twice with 10 mM Tris-HCl, pH 8.0. The cell pellet was then resuspended in a buffer containing 1.5 M LiCl, 25 mM Tris-HCl, pH 8.0, and kept on ice for 10 min. The suspension was then centrifuged, and the supernatant was collected. Proteins were precipitated using 10% Trichloroacetic acid (TCA) (w/v) at 4 °C for 1 h. The suspension was centrifuged (15,000 rpm, 30 min, 4 °C), and the pellet was washed 3 times in ice-cold Ethanol (100%), air dried, resuspended in 1× Laemmli sample buffer, and incubated at 95 °C for 10 min.

### Western blot analysis

Protein samples were separated by 4–20% Mini-PROTEAN TGX Stain-Free Precast Gels (Bio-rad), and electroblotted onto a Nitrocellulose membrane with Trans-Blot Turbo Transfer System (Bio-rad). The membrane was blocked for 1 h with 5% skim milk in TBST (50 mM Tris-Cl, pH 7.5, 150 mM NaCl, 1% Tween-20). Blots were then probed with Rabbit anti-HA or Mouse anti-GFP antibodies (1:5000 in TBST, Thermo Fisher Scientific), followed by Goat anti-rabbit fluorophore-conjugated secondary antibody StarBright Blue 700 or Goat anti-Mouse fluorophore-conjugated secondary antibody StarBright Blue 520 (Bio-Rad) (1:5000 in TBST), respectively. Gels and blots were imaged using ChemiDoc MP imaging system (Bio-Rad). A stain-Free image was used as total protein loading control.

## Quantitative reverse transcriptase PCR

RNA was extracted from *B. subtilis* cells grown to $OD_{600}$ 0.8 using Direct-zol™ RNA Miniprep Plus Kit (ZYMO RESEARCH), according to the manufacturer protocol. RNA concentration was determined using NanoDrop™ One (Thermo Scientific). RNA (2 μg) from each sample was treated with TURBO™ DNA free kit (2 Units, Thermo Fisher Scientific), and subjected to cDNA synthesis using iScript cDNA synthesis kit (Bio-Rad), according to the manufacturer protocol. qRT-PCR reactions were performed in triplicates using iTaq Universal SYBR Green Supermix (Bio-Rad), and fluorescence detection was performed using Bio-Rad CFX Connect Real-time system. RNA expression was normalized to the level of 16S rRNA. To verify that a single product was amplified, a melt curve analysis was performed using the Bio-Rad CFX Maestro software. The relative gene expression levels were calculated from threshold cycle ($C_T$) values using the $2^{-\Delta\Delta CT}$ method. qRT-PCR primers were designed using Primer3 software (available online).

## Fluorescence microscopy

Fluorescence microscopy imaging was performed using ECLIPSE Ti2 microscope (Nikon, Japan), equipped with Prime BSI camera (Photometrics, Roper Scientific, USA) or Axio Observer Z1 microscope (Zeiss). System control and image analysis were performed using NIS-Elements AR (version 5.30.07, Nikon, Japan), Metamorph 7.4 (Molecular Devices), and FIJI (Image J).

### Fluorescence microscopy of conjugation using pLS20-$P_{IPTG}$-*gfp* reporter

*B. subtilis* donor strain constitutively expressing mCherry and LacI, and harboring pLS20-$P_{IPTG}$-*gfp*, was grown to $OD_{600}$ 0.8 at 37 °C in LB medium (without IPTG), and mixed in 1:1 ratio with recipient cells grown under similar conditions. The mixture was incubated for 60–90 min at 37 °C under static conditions to permit conjugation and expression of GFP in the recipient cells. At indicated time points, samples were treated with 2% paraformaldehyde for 10 min at 25 °C, washed with 1× PBS (Phosphate Buffered Saline, pH 7.4), spotted on microscope slides (Marienfeld), covered with poly-L-lysine-coated coverslips, and visualized by fluorescence microscopy using ×100 objective. Recipient cells, expressing GFP, were considered transconjugants.

### Fluorescence microscopy of conjugation using pLS20-*ssb-yfp* reporter

*B. subtilis* donor strains harboring pLS20-*ssb-yfp*, and recipient cells constitutively expressing mCherry, were grown to $OD_{600}$ 0.8 at 37 °C in LB medium, mixed in 1:1 ratio, and incubated for 60–80 min at 37 °C under static conditions to permit conjugation and expression of SSB-YFP in the recipient cells. At indicated time points, samples were treated with 2% paraformaldehyde for 10 min at 25 °C, washed with 1× PBS (Phosphate Buffered Saline, pH 7.4), spotted on microscope slides (Marienfeld), covered with poly-L-lysine-coated coverslips, and visualized by fluorescence microscopy using 100 x objective. Recipient cells, expressing mCherry and displaying SSB-YFP foci, were considered transconjugants. For time-lapse microscopy analysis, donor and recipient strains were grown to $OD_{600}$ 0.8 at 37 °C in LB medium, mixed in 1:1 ratio, and the mixture was mounted onto a metal ring (A-7816, Invitrogen) filled with LB agarose (0.6%). Cells were incubated in a temperature and humidity-controlled chamber at 37 °C, and visualized over time by fluorescence microscopy. Conjugation efficiency was calculated as the number of transconjugants (T) /number of mCherry expressing recipients (T + R).

### Evaluating conjugative promoter activity

*B. subtilis* strains harboring $P_C$-*gfp* or $P_{33}$-*gfp* transcriptional reporters were grown to $OD_{600}$ 0.8 at 37 °C in LB medium, harvested, washed with 1× PBS, spotted on microscope slides (Marienfeld), covered with poly-L-lysine-coated coverslips and imaged by fluorescence microscopy. GFP levels for each cell were determined by processing with FIJI (Image J) "Find Maxima".

### Visualizing MC formation by time-lapse microscopy

*B. subtilis* donor cells, harboring pLS20-*ssb-yfp* in the presence or absence of the $P_{33}$-*gfp* transcriptional reporter, and recipient cells constitutively expressing mCherry, were grown to $OD_{600}$ 0.8 at 37 °C in LB medium and mixed in 1:1 ratio. The mixed cells were mounted onto a metal ring (A-7816, Invitrogen) filled with LB

agarose (0.6%), incubated in a temperature and humidity-controlled chamber (Okolab) at 37 °C for 60–80 min, and visualized by fluorescence microscopy using ×100 objective.

### Visualizing multi-species MC formation

*B. subtilis* donor cells, harboring pLS20 and expressing GFP, *B. subtilis* recipient cells constitutively expressing mCherry, and *B. megaterium*, or *B. cereus* strains, were grown to OD$_{600}$ 0.8 at 37 °C in LB medium and mixed in 1:1:1 ratio. The mixed cells were mounted onto a metal ring (A-7816, Invitrogen) filled with LB agarose (0.6%), incubated in a temperature and humidity-controlled chamber (Okolab) at 37 °C for 45 min, and visualized by fluorescence microscopy using ×100 objective.

### Flagella staining and visualization

Flagella visualization was carried out as previously described (Blair et al, 2008), with modifications. *B. subtilis* cells expressing modified flagellin (*amyE*::P$_{hag}$-*hag*$^{T209C}$) were harvested at OD$_{600}$ 0.8, washed with 1× T-BASE buffer [15 mM (NH$_4$)$_2$SO$_4$, 80 mM K$_2$HPO$_4$, 44 mM KH$_2$PO$_4$, 3.4 mM sodium citrate, and 3 mM MgSO$_4$] (pH 8), resuspended in 1× T-BASE buffer (pH 8) containing 5 µg/ml Alexa Fluor™ 594 C$_5$ Maleimide (Thermo Fisher Scientific), and incubated for 5 min at room temperature. Cells were then washed with 1× T-BASE buffer (pH 8), mounted onto a metal ring (A-7816, Invitrogen) filled with LB agarose (1.5%), and visualized by fluorescence microscopy using ×100 objective.

### Immunofluorescence

*B. subtilis* cells grown to OD$_{600}$ 0.8 at 37 °C in LB medium were harvested and washed once with 1× PBS. Cells were then treated with Rabbit anti-HA primary antibody (1:200, Thermo Fisher Scientific) for 40 min at room temperature, washed two times with 1× PBS, and subsequently treated with Donkey Anti-Rabbit Alexa647 conjugated secondary antibody (1:2000, Abcam) for 30 min at room temperature. Cells were then washed two times with 1× PBS, spotted on microscope slides (Marienfeld), covered with poly-L-lysine-coated coverslips, and visualized by fluorescence microscopy using ×100 objective.

## MC formation analysis by digital microscopy

*B. subtilis* donor strains, harboring pLS20, and recipient cells, constitutively expressing mCherry, were grown to OD$_{600}$ 0.8 at 37 °C in LB medium, mixed in 1:1 ratio, and mixtures were placed in a 96-well BLACK, CELLSTAR® plate (Greiner). Clustering of red recipient cells was followed over time at 37 °C, under static conditions, in digital wide-field microscopy (BioTek Cytation 5, Agilent). Automated Image capturing was performed at 10 min intervals using a ×10 objective and the BioTek Gen5 Software. Image processing was performed with FIJI (Image J). To estimate cluster formation (cluster analysis), images of mCherry channel were subjected to thresholding and analyzed for particles larger than 500 pixels$^2$.

## Monitoring conjugative promoter activity

*B. subtilis* cells harboring P$_C$-*gfp* or P$_{33}$-*gfp* transcriptional reporters were grown to OD$_{600}$ 0.8 at 37 °C in LB medium, harvested, washed, and resuspended in 1× PBS.

### Fluorescence of spotted cultures

Samples (10 µl) were spotted in triplicates on LB agar plates and incubated at 37 °C for 18 h. Plates were imaged using ChemiDoc MP imaging system (Bio-Rad).

### Plate reader

Samples were tested in triplicates in a 96-well BLACK, CELLSTAR® plate (Greiner). Fluorescence intensity and OD$_{600}$ were measured by Spark 10 M (Tecan) multiwell fluorometer plate reader. GFP intensities were normalized to the OD$_{600}$ values following the subtraction of the background GFP intensity of a WT donor strain (SH337, pLS20) lacking GFP.

## Statistical analysis

Statistical analysis was performed using Microsoft Office Excel software containing a data analysis add-in. To test the sample effect in grouped column data, two-way ANOVA was used. For the rest of the graphs, the Student's *t* test was calculated. *P* values obtained are indicated in the respective figure legends.

## Data availability

The source data for all microscopy images produced in this study are available in BioImage Archive (Accession: S-BIAD1375).

The source data of this paper are collected in the following database record: biostudies:S-SCDT-10_1038-S44318-024-00320-0.

## Peer review information

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

## Acknowledgements

The authors thank members of the Ben-Yehuda and Rosenshine laboratories for valuable discussions. The authors are grateful to Daniel B Kearns (Indiana U) for providing bacterial strains and anti-Hag antibodies, and Mitsuhiro Itaya (Keio U),

and David Rudner (Harvard U) for providing bacterial strains and plasmids. We thank the Hebrew University of Jerusalem, Golda Meir fellowship fund, for providing a post-doctoral fellowship to SB. This work was supported by the European Research Council (ERC) Synergy grant (810186) awarded to SB-Y and IR, and by the German Research Foundation (DFG) Priority Program SPP2389 awarded to SB-Y.

## Author contributions

**Saurabh Bhattacharya**: Conceptualization; Resources; Data curation; Formal analysis; Supervision; Validation; Investigation; Visualization; Methodology; Writing—original draft; Writing—review and editing. **Michal Bejerano-Sagie**: Conceptualization; Resources; Data curation; Formal analysis; Supervision; Validation; Investigation; Visualization; Methodology; Writing—original draft; Writing—review and editing. **Miriam Ravins**: Conceptualization; Resources; Data curation; Formal analysis; Validation; Investigation; Visualization; Methodology; Project administration; Writing—review and editing. **Liat Zeroni**: Resources; Data curation; Validation; Investigation; Visualization; Methodology. **Prabhjot Kaur**: Resources; Methodology. **Venkadesaperumal Gopu**: Resources; Methodology. **Ilan Rosenshine**: Conceptualization; Resources; Formal analysis; Supervision; Funding acquisition; Validation; Investigation; Methodology; Writing—original draft; Project administration; Writing—review and editing. **Sigal Ben-Yehuda**: Conceptualization; Resources; Formal analysis; Supervision; Funding acquisition; Validation; Investigation; Methodology; Writing—original draft; Project administration; Writing—review and editing.

Source data underlying figure panels in this paper may have individual authorship assigned. Where available, figure panel/source data authorship is listed in the following database record: biostudies:S-SCDT-10_1038-S44318-024-00320-0.

## Disclosure and competing interests statement

The authors declare no competing interests.

# Expanded View Figures

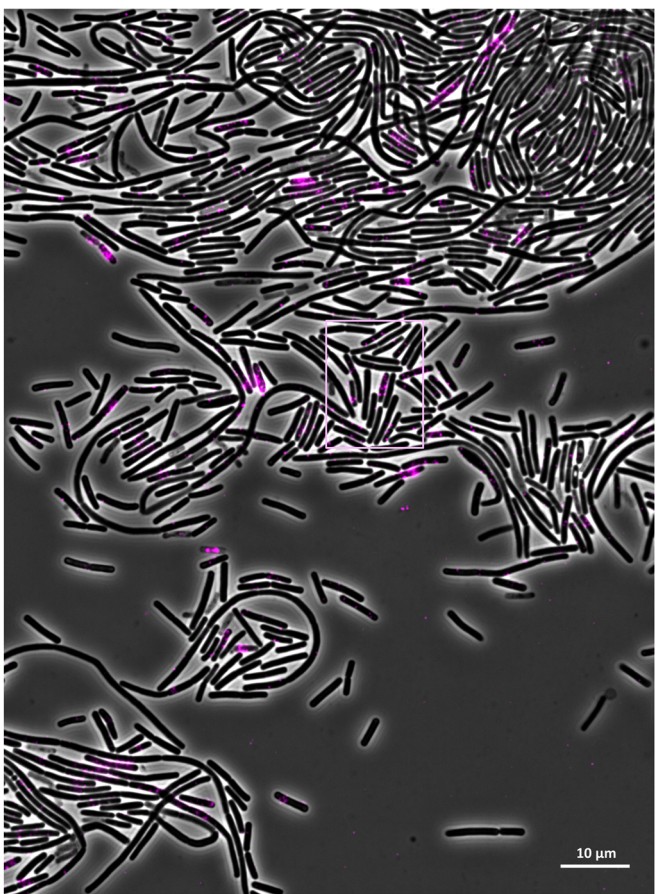

**Figure EV1.  Tie distribution on the surface of donor cells is heterogeneous.**

Image of a large field corresponding to Fig. 5D (inset). Donor cells (SH461: WT/pLS20$_{cm}$-*tie*$_{-2xHA}$) were grown to OD$_{600}$ 0.8 and visualized by immunofluorescence microscopy using primary anti-HA antibodies and Alexa647 conjugated secondary antibodies. Cells were not permeabilized before antibody treatment to enable the selective visualization of surface exposed Tie. Shown is an overlay image of phase contrast (gray) with fluorescence from Tie$_{-2xHA}$ (magenta).

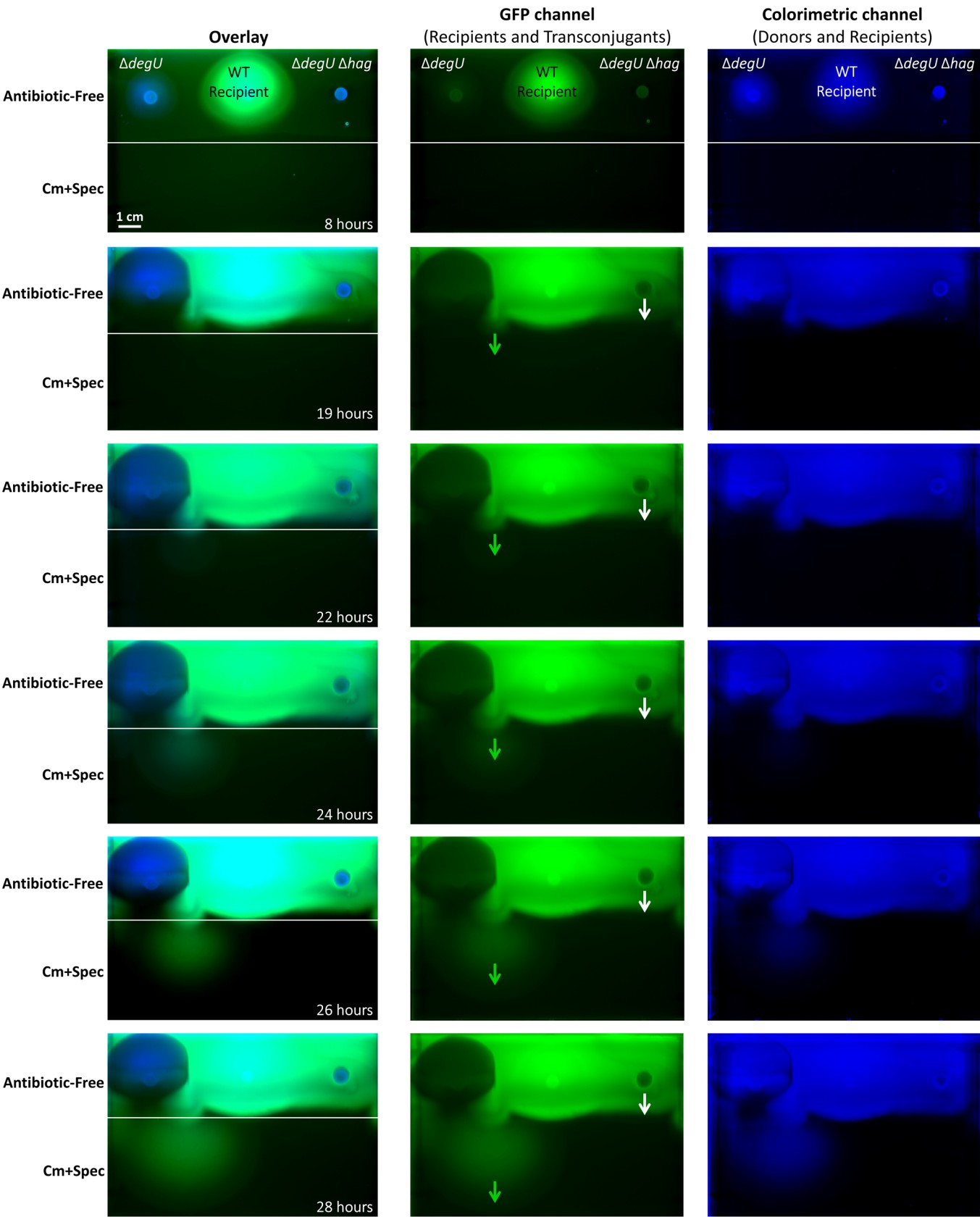

◀  **Figure EV2. Coupling motility and conjugation is beneficial for invading new niches.**

Images of entire plates corresponding to the series presented in Fig. 7B (left panels). Bacteria were grown to mid-logarithmic phase and spotted on the antibiotic-free region of the plate. Plates were spotted with Δ*degU* (SH592: Δ*degU*/pLS20$_{cm}$), Δ*hag* Δ*degU* (SH593: Δ*hag* Δ*degU*/pLS20$_{cm}$), and WT (AR16: P$_{rrnE}$-*gfp*) strains. Presented are overlay images of colorimetric channel (blue) with fluorescence from GFP (green), and their corresponding separate channels, captured at the indicated time points post incubation using ChemiDoc MP imaging system. Shown are full-plate images, including both the antibiotic-free and antibiotic-containing regions (Cm+Spec), with white lines demarcating the border between the two. Arrows indicate the intersection regions of the recipients with the motile (green) and non-motile (white) donors, and the subsequent migration of transconjugants into the selective zone.

