## [Peer Review File · The EMBO Journal]

Flagellar rotation facilitates the transfer of a bacterial conjugative plasmid

Saurabh Bhattacharya, Michal Bejerano-Sagie, Miriam Ravins, Liat Zeroni, Prabhjot Kaur, Venkadesaperumal Gopu, Ilan Rosenshine and Sigal Ben-Yehuda

Corresponding authors: Sigal Ben-Yehuda (sigalb@ekmd.huji.ac.il) , Ilan Rosenshine (ilanr@ekmd.huji.ac.il)

Review Timeline:

Submission Date:	1st Jul 24
Editorial Decision:	16th Aug 24
Revision Received:	30th Sep 24
Editorial Decision:	25th Oct 24
Revision Received:	30th Oct 24
Accepted:	7th Nov 24

Editor: Ieva Gailite

Transaction Report:

Dear Sigal,

Thank you for submitting your manuscript for consideration by the EMBO Journal. I sincerely apologise for the protracted assessment process due to the high number of manuscripts that we receive at the moment. We have now received comments from two reviewers, which are included below for your information.

As you will see from the reports, both reviewers find the study per se of interest, while also raising a number of substantive concerns regarding the experimental approach and data analysis/interpretation that would need to be addressed before they can support acceptance of the study. In particular, reviewer #1 finds that the analysis of conjugation efficiency needs to be clarified and tested with alternative methods, and the role of Tie and potential alternative factors requires further investigation. Reviewer #2 indicates that the analysis of cluster formation needs better clarification and its relevance for conjugation events should be clarified. This reviewer also requests further insight into whether inclusion of heterologous *Bacillus* species in the clusters involves DNA transfer and raises a concern with the reported hypermotility of the flgM mutant. Since I was not able to obtain the third report, I reached out to an additional advisor, who had concerns with the use of SSB-YFP as a conjugation readout and was concerned with the experimental design reported in figure 7, as only bacteria carrying the antibiotic-encoding plasmid will as such be able to invade the selective zone, thus making it difficult to disentangle the role of motility vs antibiotic resistance.

Based on the interest expressed in the referee reports, I invite you to address these concerns in a revised version of the manuscript. However, due to the substantive nature of these concerns, I think it would be helpful to discuss the revision in more detail via email or phone/videoconferencing - please let me know which option you prefer. I should also add that it is The EMBO Journal policy to allow only a single major round of revision and that it is therefore important to resolve the main concerns at this stage.

We generally allow three months as standard revision time, which can be extended to six months in the case of major revisions. Should you foresee a problem in meeting this deadline, please let us know in advance to discuss an extension. As a matter of policy, competing manuscripts published during this period will not negatively impact on our assessment of the conceptual advance presented by your study. However, please contact me as soon as possible upon publication of any related work to discuss the appropriate course of action.

When preparing your letter of response to the referees' comments, please bear in mind that this will form part of the Review Process File and will therefore be available online to the community. For more details on our Transparent Editorial Process, please visit our website: <https://www.embopress.org/page/journal/14602075/authorguide#transparentprocess>. Please also see the attached instructions for further guidelines on preparation of the revised manuscript.

Please feel free to contact me if you have any further questions regarding the revision. Thank you for the opportunity to consider your work for publication. I look forward to discussing your revision.

With best regards,

Ieva

Ieva Gailite, PhD
Senior Scientific Editor
The EMBO Journal
Meyerohofstrasse 1
D-69117 Heidelberg
Tel: +4962218891309

- a point-by-point response to the referees' comments, with a detailed description of the changes made (as a word file).
- a word file of the manuscript text.

- individual production quality figure files (one file per figure)

- a complete author checklist, which you can download from our author guidelines

(<https://www.embopress.org/page/journal/14602075/authorguide>).

- Expanded View files (replacing Supplementary Information)

- a Reagents and Tools Table as part of the Methods section, which can be downloaded from our author guidelines

(<https://www.embopress.org/page/journal/14602075/authorguide#structuredmethods>)

We realize that it is difficult to revise to a specific deadline. In the interest of protecting the conceptual advance provided by the work, we recommend a revision within 3 months (14th Nov 2024). Please discuss the revision progress ahead of this time with the editor if you require more time to complete the revisions.

Referee #1:

The study by Bhattacharya, Bejerano-Sagie, and colleagues investigates the role of flagellar-mediated motility in the conjugation of the Bacillus-specific plasmid pLS20, aiming to understand why conjugation in liquid medium is preferred over solid surfaces.

The authors employ time-lapse fluorescence microscopy to demonstrate that strains carrying the conjugative plasmid facilitate the formation of mating clusters, which act as hot spots for pLS20 conjugation. While mating cluster formation was not caused by the assembly of flagella on the surface of donor or recipient cells, the assembly of functional flagella was essential for aggregate formation in a pLS20-dependent manner. By examining the expression of essential conjugative genes in different flagellar gene mutants, the authors identified a previously unidentified promoter driving the expression of a subset of genes essential for conjugation. The presence of functional flagella increases the expression of these genes, leading to a dependency on flagella for the conjugation of pLS20. Additionally, the authors demonstrate that flagella function as mechanosensors, controlling gene expression under liquid conditions through the action of a two-component system, DegS-DegU. Finally, they show how motility and conjugation enable the colonization of new environments where the acquisition of conjugative plasmids is necessary for colonization.

Overall, the paper conclusively demonstrates the relationship between motility and the conjugation of pLS20 in Bacilli, providing relevant evidence for the motility-dependency of DNA transfer in liquid medium.

Comments:

1) Conjugation Assays by CFU: It is unclear how the authors have determined the conjugation efficiency by CFU. Due to the high number of colonies formed on the spots, individual colonies cannot be discerned, which may obscure relevant information.

For example, differences in the conjugation efficiency for flagellar gene mutants in the overexpressing RapA background (Figure 3A) or in the flagellar recipient mutants (Figure 2F). Could the authors clarify the method used to accurately determine conjugation efficiency in these densely populated spots? Detailed descriptions or alternative methods to distinguish individual colonies could enhance the reliability of the results.

2) For all figures analyzing conjugation efficiency, please specify that this was determined by the number of transconjugants per number of recipients (T/T+R) in the axis titles.

3) All figures: The choice of the different performed statistical tests is unclear. Further, the number of replicates should be indicated in the individual figure panels. Instead of bar graphs, individual data points should be shown.

4) Figure 5: The deletion of tie leads to a complete abolishment of conjugation, while mating cluster formation is only slightly affected in the absence of the adhesin. This suggests the existence of other components involved in the formation of mating clusters and raises questions about the specific mechanisms through which Tie contributes to conjugation efficiency. Are there any other described mechanisms by which Tie modulates conjugation? Alternatively, are there other proteins involved in the stabilization of mating pairs?

5) Figures 4 and 5: Figure 4 demonstrates that expression from the PC and P33 promoters is quite heterologous and rather rare within the population. However, Figure 5 shows a relatively homogeneous decoration of the population with Tie-2xHA. Could the authors elaborate on the mechanisms leading to this apparent discrepancy?

6) Figure 5C: Was any control for the detection of cytosolic proteins conducted? It is required to confirm that the observed signal arises solely from membrane-bound proteins and does not result from the overcarriage of proteins from the cytosolic part of the sample. This control is essential for validating the specificity of the detected signals.

7) Figure 6A: Regarding the hyperconjugative strain overexpressing RapA, is it still capable of conjugating the plasmid (pSL20-Cm)? Furthermore, could the conjugation of the plasmid into plasmid-naïve cells inhibit the acquisition of the plasmid from Δ hag / pLS20 strains through surface or entry exclusion systems? This potential exclusion could have implications in the transfer of the plasmid from the Δ hag in the mixed culture.

8) Page 11, line 3, "Considering that conjugation is a relatively rare event ($\sim 2.5 \times 10^{-3}$)," it would be helpful to specify how this conjugation efficiency was determined-i.e., whether it is based on T/T+R or T/D.

9) Page 33, line 14, remove the comment "(should we quantify using an area threshold for phase?)"

Referee #2:

In this work, the authors show that activation of critical parts of the pLS20 encoded conjugal transfer system in *B. subtilis* requires the response regulator DegU. Activity of DegU has been linked to flagellar function, and most of the paper is the detective work to lead them to the flagellar motor and DegU regulation. They show that DegU activates the conjugal transfer system in a small subpopulation of cells, around which recipients aggregate and with a clever SSB-YFP reporter, can infer genetic transfer events. In sum the work explains previous observations that cell motility was required for transfer, and further shows that motility not only regulates transfer proteins in the donor, but is also required to disperse the donor as individuals for high efficiency transfer. The biggest problem I had with the work was in the wide variety of different assays used to draw conclusions and the difficulty comparing the assays between experiments and figures. I think all of the data lines up but what is needed is a master figure with all of the mutants using conjugant frequency (which is slightly different than the conjugation efficiency of SSB-YFP and has 7 orders of magnitude dynamic range) like that shown in Fig 3A. This allows immediate and powerful comparison of the different mutant phenotypes, and the assay is simple rapid and comparable between labs.

p. 6 line 16. Clarify "semi-liquid"

Fig 1C. The clusters are very difficult to see and I didn't quite understand what was being indicated in the "cluster analysis" panel series. Maybe it was because the clusters in 1C looked so different than the zooms in 1B or maybe 1C just needs better explanation in text.

p.7 line 18. While unrelated species cluster with a *B. subtilis* pLS20 donor, was there any indication of SSB-YFP transfer to *B. cereus* or *B. megaterium*? I presume not but maybe pLS20 genes aren't expressed in those organisms. Whatever the case, I feel there should be mention of whether or not this happened and perhaps some explanation as to why or why not.

P.7 line 21. To the best of my knowledge, a flgM mutant is not hyperflagellated in *B. subtilis* or other bacteria. Mutation of FlgM increases SigD/Sig28 dependent gene expression but these are late class flagellar genes. P. 8 line 18. I also don't believe that a flgM mutant has been reported to be hypermotile. While not quantitative, the assay in Fig 2D might suggest that the flgM mutant

is more motile than the wild type but I'm not sure why or how that would be. Maybe motAB overexpression?

Fig 2EF. What is the Y-axis on these graphs? Is it linear, or log? It looks like neither. I believe the data was generated by the frequency of antibiotic resistance gene transfer so the dynamic range should be much larger than what is shown here. What was the lower limit of detection? 10^{-7} ? 10^{-8} ? On a log scale that goes to lower limit, the effects of mutants like hag or sigD could look massive, whereas the effects of mutants like motA or flgM might look to be barely detectable. In short, could these results be projected on a graph similar to that of Fig 3A?

p. 18 line 3. "Our findings unravel the riddle of efficient conjugation of pLS20 in liquid medium despite lacking an apparent pilus" Clarify. I feel like the work aids our understanding of why motility is required but I don't see how it explains the lack of pilus.

As you will see from the reports, both reviewers find the study per se of interest, while also raising a number of substantive concerns regarding the experimental approach and data analysis/interpretation that would need to be addressed before they can support acceptance of the study. In particular, reviewer #1 finds that the analysis of conjugation efficiency needs to be clarified and tested with alternative methods, and the role of Tie and potential alternative factors requires further investigation. Reviewer #2 indicates that the analysis of cluster formation needs better clarification and its relevance for conjugation events should be clarified. This reviewer also requests further insight into whether inclusion of heterologous *Bacillus* species in the clusters involves DNA transfer and raises a concern with the reported hypermotility of the *flgM* mutant.

Please see below our point-by-point response to all the Reviewers' concerns.

Since I was not able to obtain the third report, I reached out to an additional advisor, who had concerns with the use of SSB-YFP as a conjugation readout,

We found the SSB-YFP fusion to be an efficient, reliable and accurate method for measuring conjugation events for the following reasons:

- 1) The native *ssb* gene on pLS20 was fused to YFP, resulting in a fully functional protein. When transferred to recipient cells, which are mCherry-labeled, bright and prominent SSB-YFP foci emerge in transconjugant red bacteria (Figure 1A). These foci can only appear after receiving the plasmid DNA.
- 2) SSB is one of the earliest proteins to be robustly expressed in transconjugants (Couturier et al., 2023; Masai & Arai, 1997).
- 3) We compared the *ssb-yfp* reporter with another conjugation reporter we constructed, based on *gfp*, which was inserted into pLS20 (Figure S1A-S1B). The expression of this *gfp* reporter is repressed in the donor by *Lacl* and is derepressed in the recipient, which lacks *Lacl*. We found that SSB-YFP was expressed earlier in the transconjugants compared to the GFP reporter, with both reporters exhibiting similar conjugation frequencies.
- 4) Conjugation efficiencies calculated for different mutants using the *ssb-yfp* reporter were consistent with those obtained from CFU analysis (Figures 2B and 2E).

This will be better explained in the revised version.

and was concerned with the experimental design reported in figure 7, as only bacteria carrying the antibiotic-encoding plasmid will as such be able to invade the selective zone, thus making it difficult to disentangle the role of motility vs antibiotic resistance.

The plasmid can be considered as an evolutionary unit. The rationale behind the experimental design in Figure 7 was to demonstrate the advantage the plasmid gains by synchronizing the expression of its costly conjugation system with host motility. This synchronization is the key outcome of our study.

Testing this issue was enabled due to our discovery that the *degU* mutant uncouples conjugation gene expression from motility. We thus constructed and compared two donor strains, motile and non-motile, yet similarly expressing the conjugation system. We show that plasmid spread is much more efficient when carried by the motile strain, thus underscoring the advantage of linking conjugation gene expression with host motility.

This will be further explained and clarified in the revised version.

Referee #1:

The study by Bhattacharya, Bejerano-Sagie, and colleagues investigates the role of flagellar-mediated motility in the conjugation of the Bacillus-specific plasmid pLS20, aiming to understand why conjugation in liquid medium is preferred over solid surfaces.

The authors employ time-lapse fluorescence microscopy to demonstrate that strains carrying the conjugative plasmid facilitate the formation of mating clusters, which act as hot spots for pLS20 conjugation. While mating cluster formation was not caused by the assembly of flagella on the surface of donor or recipient cells, the assembly of functional flagella was essential for aggregate formation in a pLS20-dependent manner. By examining the expression of essential conjugative genes in different flagellar gene mutants, the authors identified a previously unidentified promoter driving the expression of a subset of genes essential for conjugation. The presence of functional flagella increases the expression of these genes, leading to a dependency on flagella for the conjugation of pLS20. Additionally, the authors demonstrate that flagella function as mechanosensors, controlling gene expression under liquid conditions through the action of a two-component system, DegS-DegU. Finally, they show how motility and conjugation enable the colonization of new environments where the acquisition of conjugative plasmids is necessary for colonization.

Overall, the paper conclusively demonstrates the relationship between motility and the conjugation of pLS20 in Bacilli, providing relevant evidence for the motility-dependency of DNA transfer in liquid medium.

Comments:

1) Conjugation Assays by CFU: It is unclear how the authors have determined the conjugation efficiency by CFU. Due to the high number of colonies formed on the spots, individual colonies cannot be discerned, which may obscure relevant information. For example, differences in the conjugation efficiency for flagellar gene mutants in the overexpressing RapA background (Figure 3A) or in the flagellar recipient mutants (Figure 2F). Could the authors clarify the method used to accurately determine conjugation efficiency in these densely populated spots? Detailed descriptions or alternative methods to distinguish individual colonies could enhance the reliability of the results.

The spots presented in the Figures throughout the manuscript are for visual representation only. For each experiment, quantification was performed using serial dilutions that allowed for accurate counting of well-separated colonies. This will be better described in the revised version. If needed, the raw data will be deposited with the manuscript to further address this point.

2) For all figures analyzing conjugation efficiency, please specify that this was determined by the number of transconjugants per number of recipients (T/T+R) in the axis titles.

This will be done in the revised manuscript.

3) All figures: The choice of the different performed statistical tests is unclear. Further, the number of replicates should be indicated in the individual figure panels. Instead of bar graphs, individual data points should be shown.

We will clarify our choice of statistical tests and make adjustments if necessary. Individual data points will be shown in the bar graphs.

4) Figure 5: The deletion of tie leads to a complete abolishment of conjugation, while mating cluster formation is only slightly affected in the absence of the adhesin. This suggests the existence of other components involved in the formation of mating clusters and raises questions about the specific mechanisms through which Tie contributes to conjugation efficiency. Are there any other described mechanisms by which Tie modulates conjugation?

Tie has been reported to act as an adhesion protein (Gago-Córdoba et al., 2021). In contrast, our results show relatively a minor role for Tie in cluster formation. Furthermore, we found Tie to be essential for conjugation, whereas adhesion proteins typically only partially affect the process. Other mechanisms for Tie modulating conjugation have not been described so far and remain to be elucidated.

Alternatively, are there other proteins involved in the stabilization of mating pairs?

Using bioinformatics, we could not identify so far a potential candidate protein to be involved in the stabilization of mating pairs.

5) Figures 4 and 5: Figure 4 demonstrates that expression from the PC and P33 promoters is quite heterologous and rather rare within the population. However, Figure 5 shows a relatively homogeneous decoration of the population with Tie-2xHA. Could the authors elaborate on the mechanisms leading to this apparent discrepancy?

In fact, we do see a variation in Tie distribution over the donor surface. We will add a quantification analysis to the manuscript and discuss it in detail.

6) Figure 5C: Was any control for the detection of cytosolic proteins conducted? It is required to confirm that the observed signal arises solely from membrane-bound proteins and does not result from the overcarriage of proteins from the cytosolic part

of the sample. This control is essential for validating the specificity of the detected signals.

This experiment will be repeated with the requested control.

7) Figure 6A: Regarding the hyperconjugative strain overexpressing RapA, is it still capable of conjugating the plasmid (pSL20-Cm)?

Yes, it is capable of transferring pSL20-Cm. We will include this data in the revised version.

Furthermore, could the conjugation of the plasmid into plasmid-naïve cells inhibit the acquisition of the plasmid from Δhag / pLS20 strains through surface or entry exclusion systems? This potential exclusion could have implications in the transfer of the plasmid from the Δhag in the mixed culture.

The recipient cells are in large excess relative to the low conjugation rate. Therefore, we do not expect inhibition by exclusion mechanisms. The conjugation efficiency of a strain overexpressing RapA is approximately 10^{-2} (Figure 3A), indicating that only 1% of the population could restrict plasmid acquisition from Δhag donor through exclusion systems. Nevertheless, to address the Reviewer's concern, we will repeat the experiment using a WT donor (instead of Δhag) mixed with a donor overexpressing RapA as an additional control.

8) Page 11, line 3, "Considering that conjugation is a relatively rare event ($\sim 2.5 \times 10^{-3}$)," it would be helpful to specify how this conjugation efficiency was determined-i.e., whether it is based on T/T+R or T/D.

As requested, we will specify how conjugation efficiency was determined.

9) Page 33, line 14, remove the comment "(should we quantify using an area threshold for phase?)"

Thank you, this will be edited.

Referee #2:

In this work, the authors show that activation of critical parts of the pLS20 encoded conjugal transfer system in *B. subtilis* requires the response regulator DegU. Activity of DegU has been linked to flagellar function, and most of the paper is the detective work to lead them to the flagellar motor and DegU regulation. They show that DegU activates the conjugal transfer system in a small subpopulation of cells, around which recipients aggregate and with a clever SSB-YFP reporter, can infer genetic transfer events. In sum the work explains previous observations that cell motility was required for transfer, and further shows that motility not only regulates transfer proteins in the donor, but is also required to disperse the donor as individuals for high efficiency transfer. The biggest problem I had with the work was in the wide variety of different assays used to draw conclusions and the difficulty comparing the assays between experiments and figures. I think all of the data lines up but what is needed is a master figure with all of the mutants using conjugant frequency (which is slightly different than the conjugation efficiency of SSB-YFP and has 7 orders of magnitude dynamic range) like that shown in Fig 3A. This allows immediate and powerful comparison of the different mutant phenotypes, and the assay is simple rapid and comparable between labs.

We will add graphs with the actual conjugation efficiencies throughout the manuscript.

p. 6 line 16. Clarify "semi-liquid"

We used 0.6% agarose, and this will be better indicated in the manuscript.

Fig 1C. The clusters are very difficult to see and I didn't quite understand what was being indicated in the "cluster analysis" panel series. Maybe it was because the clusters in 1C looked so different than the zooms in 1B or maybe 1C just needs better explanation in text.

The cluster analysis will be better explained in the revised manuscript. The methodologies used in Figure 1B and 1C differ: 1B was captured at high magnification, zooming into the formation of a single cluster over time, whereas 1C was captured at low magnification to quantify cluster formation across large fields. This distinction will be emphasized in the revised manuscript.

p.7 line 18. While unrelated species cluster with a *B. subtilis* pLS20 donor, was there any indication of SSB-YFP transfer to *B. cereus* or *B. megaterium*? I presume not but maybe pLS20 genes aren't expressed in those organisms. Whatever the case, I feel there should be mention of whether or not this happened and perhaps some explanation as to why or why not.

Although pLS20 is widespread in *Bacilli*, we were unable to monitor the transfer of pLS20 to *B. cereus* or *B. megaterium* using the SSB-YFP reporter. This observation is in

line with previous reports where direct transfer of pLS20 to these species was not observed, but pLS20-mediated transfer of other mobilizable plasmids was detected (Koehler & Thorne, 1987), thus, highlighting the importance of pLS20-induced mating clusters in nature.

This will be discussed in the revised manuscript.

P.7 line 21. To the best of my knowledge, a *flgM* mutant is not hyperflagellated in *B. subtilis* or other bacteria. Mutation of *FlgM* increases SigD/Sig28 dependent gene expression but these are late class flagellar genes. P. 8 line 18. I also don't believe that a *flgM* mutant has been reported to be hypermotile. While not quantitative, the assay in Fig 2D might suggest that the *flgM* mutant is more motile than the wild type but I'm not sure why or how that would be. Maybe *motAB* overexpression?

In *B. subtilis*, it has been reported that the influence of SigD is not limited to late-class flagellar genes, but also extends to the regulation of other flagellar components including flagellar hook, basal body, and export apparatus (Mukherjee & Kearns, 2014). In agreement, we have evidence that there are more flagellated motile cells in the *flgM* mutant population in comparison to WT population, as supported by the motility assay in Fig 2D. We will include a quantitative analysis to substantiate this issue in the revised version.

Fig 2EF. What is the Y-axis on these graphs? Is it linear, or log? It looks like neither. I believe the data was generated by the frequency of antibiotic resistance gene transfer so the dynamic range should be much larger than what is shown here. What was the lower limit of detection? 10^{-7} ? 10^{-8} ? On a log scale that goes to lower limit, the effects of mutants like *hag* or *sigD* could look massive, whereas the effects of mutants like *motA* or *flgM* might look to be barely detectable. In short, could these results be projected on a graph similar to that of Fig 3A?

We will adjust the Y axis as requested.

p. 18 line 3. "Our findings unravel the riddle of efficient conjugation of pLS20 in liquid medium despite lacking an apparent pilus" Clarify. I feel like the work aids our understanding of why motility is required but I don't see how it explains the lack of pilus.

The reviewer is correct. This will be modified.

References

- Couturier, A., Virolle, C., Goldlust, K., Berne-Dedieu, A., Reuter, A., Nolivos, S., Yamaichi, Y., Bigot, S., & Lesterlin, C. (2023). Real-time visualisation of the intracellular dynamics of conjugative plasmid transfer. *Nature Communications*, *14*(1), 294. <https://doi.org/10.1038/s41467-023-35978-3>
- Gago-Córdoba, C., Val-Calvo, J., Abia, D., Díaz-Talavera, A., Miguel-Arribas, A., Aguilar Suárez, R., van Dijl, J. M., Wu, L. J., & Meijer, W. J. J. (2021). A conserved class II type thioester domain-containing adhesin is required for efficient conjugation in *Bacillus subtilis*. *MBio*, *12*(2). <https://doi.org/10.1128/mBio.00104-21>
- Koehler, T. M., & Thorne, C. B. (1987). *Bacillus subtilis* (*natto*) plasmid pLS20 mediates interspecies plasmid transfer. *Journal of Bacteriology*, *169*(11), 5271–5278. <https://doi.org/10.1128/jb.169.11.5271-5278.1987>
- Masai, H., & Arai, K. (1997). Frpo: a novel single-stranded DNA promoter for transcription and for primer RNA synthesis of DNA replication. *Cell*, *89*(6), 897–907. [https://doi.org/10.1016/s0092-8674\(00\)80275-5](https://doi.org/10.1016/s0092-8674(00)80275-5)
- Mukherjee, S., & Kearns, D. B. (2014). The structure and regulation of flagella in *Bacillus subtilis*. *Annual Review of Genetics*, *48*(1), 319–340. <https://doi.org/10.1146/annurev-genet-120213-092406>

We would like to thank the Referees for the helpful and constructive comments on our manuscript: "**A conjugative plasmid exploits flagella rotation as a cue to facilitate its transfer**" (EMBOJ-2024-118350). In the revised manuscript, we attempted to address all the concerns raised by the Referees. Below we provide a summary of our major modifications, followed by a detailed point-by-point response.

Summary of the major changes

1) Conjugation efficiencies are now plotted on the same scale to include a larger dynamic range, enabling the direct comparison between all the experiments presented throughout the manuscript (e.g., Fig. 2E-2F, Fig. 6J).

2) We clarified our choice of statistical tests in the Methods and Protocols section, and we now indicate the number of replicates for each figure panel in the figure legends. Individual data points, indicative of the number of replicates, have now been included for the bar graphs (i.e., Fig. 1D, 1F; Fig. 3C-3D; Fig. 4B; Fig. 5E-5F).

3) We conducted additional experiments substantiating that flagella activity is required *in cis* by the donor, and cannot be complemented in trans (Fig. 6A-6B).

4) Additional data on Tie distribution over the donor surface have been included (Fig. 5E; Fig. EV1; Appendix Fig. S2D).

As you will see from the reports, both reviewers find the study per se of interest, while also raising a number of substantive concerns regarding the experimental approach and data analysis/interpretation that would need to be addressed before they can support acceptance of the study. In particular, reviewer #1 finds that the analysis of conjugation efficiency needs to be clarified and tested with alternative methods, and the role of Tie and potential alternative factors requires further investigation. Reviewer #2 indicates that the analysis of cluster formation needs better clarification and its relevance for conjugation events should be clarified. This reviewer also requests further insight into whether inclusion of heterologous *Bacillus* species in the clusters involves DNA transfer and raises a concern with the reported hypermotility of the flgM mutant.

Since I was not able to obtain the third report, I reached out to an additional advisor, who had concerns with the use of SSB-YFP as a conjugation readout,

To address this Reviewer's concern, we would like to clarify the decision to employ the SSB-YFP fusion as a readout for conjugation.

- 1) SSB is one of the earliest proteins to be robustly expressed in transconjugants (Couturier *et al.*, 2023; Masai & Arai, 1997). Therefore, the native *ssb* gene on pLS20 was fused to YFP, resulting in a fully functional protein. When transferred to recipient cells, which are mCherry-labeled, bright and prominent SSB-YFP foci emerge in transconjugant red bacteria (Fig. 1A). These foci only appeared after receiving the plasmid DNA.
- 2) We compared the *ssb-yfp* reporter with another conjugation reporter we constructed, based on *gfp*, which was inserted into pLS20 (p6 L5-15; Appendix Fig. S1A-S1B). The expression of this *gfp* reporter is repressed in the donor by LacI and is derepressed in the recipient, which lacks LacI. We showed that SSB-YFP was expressed earlier in the transconjugants compared to the GFP reporter, with both reporters exhibiting similar conjugation frequencies.
- 3) Conjugation efficiencies calculated for different mutants using the *ssb-yfp* reporter were consistent with those obtained from CFU analysis (Fig. 2B, 2E).

Altogether, we found the *ssb-yfp* reporter to be an efficient, reliable, and accurate method for measuring conjugation events.

and was concerned with the experimental design reported in figure 7, as only bacteria carrying the antibiotic-encoding plasmid will as such be able to invade the selective zone, thus making it difficult to disentangle the role of motility vs antibiotic resistance.

The plasmid can be considered as an evolutionary unit. The rationale behind the experimental design in Fig. 7 was to demonstrate the advantage the plasmid gains by synchronizing the expression of its costly conjugation system with host motility. This synchronization is the key outcome of our study.

Testing this issue was enabled due to our findings that the *degU* mutant uncouples conjugation gene expression from motility. We thus constructed and compared two donor strains, motile and non-motile, yet similarly expressing the conjugation system. We show that plasmid spread is much more efficient when carried by the motile strain, thus underscoring the advantage of linking conjugation gene expression with host motility. To emphasize this point, we included in the revised version data showing that both donors intersect with the motile recipient bacteria (Fig. EV2). However, as this encounter occurs earlier when the donor is motile, transconjugants that received the plasmid from the motile donor invade the selective area more rapidly (Fig. 7A-7B; Fig. EV2). This is now better clarified in the revised version (p14 L6-22).

Referee #1:

The study by Bhattacharya, Bejerano-Sagie, and colleagues investigates the role of flagellar-mediated motility in the conjugation of the Bacillus-specific plasmid pLS20, aiming to understand why conjugation in liquid medium is preferred over solid surfaces.

The authors employ time-lapse fluorescence microscopy to demonstrate that strains carrying the conjugative plasmid facilitate the formation of mating clusters, which act as hot spots for pLS20 conjugation. While mating cluster formation was not caused by the assembly of flagella on the surface of donor or recipient cells, the assembly of functional flagella was essential for aggregate formation in a pLS20-dependent manner. By examining the expression of essential conjugative genes in different flagellar gene mutants, the authors identified a previously unidentified promoter driving the expression of a subset of genes essential for conjugation. The presence of functional flagella increases the expression of these genes, leading to a dependency on flagella for the conjugation of pLS20. Additionally, the authors demonstrate that flagella function as mechanosensors, controlling gene expression under liquid conditions through the action of a two-component system, DegS-DegU. Finally, they show how motility and conjugation enable the colonization of new environments where the acquisition of conjugative plasmids is necessary for colonization.

Overall, the paper conclusively demonstrates the relationship between motility and the conjugation of pLS20 in Bacilli, providing relevant evidence for the motility-dependency of DNA transfer in liquid medium.

Comments:

1) Conjugation Assays by CFU: It is unclear how the authors have determined the conjugation efficiency by CFU. Due to the high number of colonies formed on the spots, individual colonies cannot be discerned, which may obscure relevant information. For example, differences in the conjugation efficiency for flagellar gene mutants in the overexpressing RapA background (Figure 3A) or in the flagellar recipient mutants (Figure 2F). Could the authors clarify the method used to accurately

determine conjugation efficiency in these densely populated spots? Detailed descriptions or alternative methods to distinguish individual colonies could enhance the reliability of the results.

The representative spots displayed in the Figures throughout the manuscript are for visual representation only. For each experiment, quantification was performed using serial dilutions that allowed for accurate counting of well-separated colonies. This is now better explained in the Methods and Protocols section of the revised manuscript (p19 L20-p20 L5). Furthermore, the CFU-based conjugation efficiencies are now plotted on the same scale, enabling the direct comparison between all experiments presented along the manuscript (e.g. Fig. 2E-2F, Fig. 6J).

2) For all figures analyzing conjugation efficiency, please specify that this was determined by the number of transconjugants per number of recipients (T/T+R) in the axis titles.

As requested by the Reviewer, conjugation efficiency is now presented as the number of transconjugants per number of total recipients (T/T+R) in the axis titles, figure legends (e.g., Fig. 2E-F; Fig. 3A; Fig. 6B, 6J), and explained in the Methods and Protocols section of the revised manuscript (p19 L20-p20 L5).

3) All figures: The choice of the different performed statistical tests is unclear. Further, the number of replicates should be indicated in the individual figure panels. Instead of bar graphs, individual data points should be shown.

To test the sample effect in grouped column data, two-way ANOVA was performed. For the rest of the graphs, Student's T-Test was performed. This is now indicated in the figure legends of the revised manuscript, and explained in the Methods and Protocols section (p26 L14-17).

The number of replicates for each figure panel is indicated in the respective figure legends.

Individual data points, indicative of the number of replicates, have now been included for the bar graphs (Fig. 1D, 1F; Fig. 3C-3D; Fig. 4B; Fig. 5E-5F).

4) Figure 5: The deletion of tie leads to a complete abolishment of conjugation, while mating cluster formation is only slightly affected in the absence of the adhesin. This suggests the existence of other components involved in the formation of mating clusters and raises questions about the specific mechanisms through which Tie contributes to conjugation efficiency. Are there any other described mechanisms by which Tie modulates conjugation?

Tie has been reported to act as an adhesion protein (Gago-Córdoba et al., 2021). In contrast, our results show a relatively minor role for Tie in cluster formation. Furthermore, we found Tie to be essential for conjugation, whereas adhesion proteins typically only partially affect the process. Other mechanisms for Tie modulating conjugation have not been described so far and remain to be elucidated.

Alternatively, are there other proteins involved in the stabilization of mating pairs?

Using bioinformatics, we could not identify so far, a potential candidate protein to be involved in the stabilization of mating pairs. This demands further investigation, which we are now pursuing. The possible involvement of additional pLS20 factors in mediating mating cluster formation is now mentioned in the revised manuscript (p12 L10-13).

5) Figures 4 and 5: Figure 4 demonstrates that expression from the PC and P33 promoters is quite heterologous and rather rare within the population. However, Figure 5 shows a relatively homogeneous decoration of the population with Tie-2xHA. Could the authors elaborate on the mechanisms leading to this apparent discrepancy?

We thank the Reviewer for this comment. In fact, there is a variation in Tie distribution over the donor surface. We have now included a larger microscopy field and additional quantification data for Tie-2xHA foci distribution within the population to further substantiate this point (p12 L5-8; Fig. 5E; Fig. EV1). Our data suggests that the presence of Tie-2xHA foci is heterogeneous, with only a minor subpopulation of the cells exhibiting high numbers of foci (>3), in line with P_C and P₃₃ expression distribution.

6) Figure 5C: Was any control for the detection of cytosolic proteins conducted? It is required to confirm that the observed signal arises solely from membrane-bound proteins and does not result from the overcarriage of proteins from the cytosolic part of the sample. This control is essential for validating the specificity of the detected signals.

As requested by the Reviewer, control experiments were conducted to confirm that the signal arises specifically from the cell wall fraction. Indeed, only a negligible signal was obtained for cytosolic protein (GFP) in the cell wall fraction, highlighting the specificity of the detected signal for the cell wall (Appendix Fig. S2D).

7) Figure 6A: Regarding the hyperconjugative strain overexpressing RapA, is it still capable of conjugating the plasmid (pSL20-Cm)?

Yes, it is capable of transferring pSL20_{cm}. This data is now presented in the revised manuscript (Fig. 6B).

Furthermore, could the conjugation of the plasmid into plasmid-naïve cells inhibit the acquisition of the plasmid from Δ hag / pLS20 strains through surface or entry exclusion systems? This potential exclusion could have implications in the transfer of the plasmid from the Δ hag in the mixed culture.

To address the Reviewer's concern, we have now conducted the experiment with an additional control of a WT donor (instead of Δ hag) mixed with a donor overexpressing *rapA*. Our data indicates that conjugation efficiencies of the WT donor remained similar irrespective of the presence of the other donor, WT or *rapA* overexpression, ruling out the possibility of surface/entry exclusion in the mixed cultures. These data are now presented in the revised manuscript (Fig. 6A-6B).

8) Page 11, line 3, "Considering that conjugation is a relatively rare event ($\sim 2.5 \times 10^{-3}$)," it would be helpful to specify how this conjugation efficiency was determined-i.e., whether it is based on T/T+R or T/D.

As requested by the Reviewer, the conjugation efficiency is presented as the number of transconjugants per number of total recipients (T/T+R) and is indicated in the axis titles and the corresponding figure legends of the revised manuscript (Fig. 2E-2F; Fig. 3A; Fig. 6B, 6J).

9) Page 33, line 14, remove the comment "(should we quantify using an area threshold for phase?)"

Thank you, this comment was removed from the revised manuscript.

Referee #2:

In this work, the authors show that activation of critical parts of the pLS20 encoded conjugal transfer system in *B. subtilis* requires the response regulator DegU. Activity of DegU has been linked to flagellar function, and most of the paper is the detective work to lead them to the flagellar motor and DegU regulation. They show that DegU activates the conjugal transfer system in a small subpopulation of cells, around which recipients aggregate and with a clever SSB-YFP reporter, can infer genetic transfer events. In sum the work explains previous observations that cell motility was required for transfer, and further shows that motility not only regulates transfer proteins in the donor, but is also required to disperse the donor as individuals for high efficiency transfer. The biggest problem I had with the work was in the wide variety of different assays used to draw conclusions and the difficulty comparing the assays between experiments and figures. I think all of the data lines up but what is needed is a master figure with all of the mutants using conjugant frequency (which is slightly different than the conjugation efficiency of SSB-YFP and has 7 orders of magnitude dynamic range) like that shown in Fig 3A. This allows immediate and powerful comparison of the different mutant phenotypes, and the assay is simple rapid and comparable between labs.

We thank the Reviewer for this very helpful comment. We have now modified the Y-axis of all the graphs presenting CFU-based conjugation frequencies to include a larger dynamic range (10^{-6} to 10^{-1}), enabling the direct comparison among the experiments (e.g. Fig. 2E-2F; Fig. 6B). Furthermore, as proposed, we included an additional figure, summarizing the experiments shown in Figure 6, within the same dynamic range (Fig. 6J).

p. 6 line 16. Clarify "semi-liquid"

We used 0.6% agarose. This is now indicated in the revised manuscript (p6 L16-18).

Fig 1C. The clusters are very difficult to see and I didn't quite understand what was being indicated in the "cluster analysis" panel series. Maybe it was because the clusters in 1C looked so different than the zooms in 1B or maybe 1C just needs better explanation in text.

The methodologies used in Figure 1B and 1C differ: 1B was captured at high magnification, zooming into the formation of a single cluster over time under semi-liquid conditions, whereas 1C was captured at low magnification to quantify cluster formation across large fields under liquid conditions. This distinction is now emphasized in the revised manuscript in the main text (p7 L3-6), and Methods and Protocols section (p24 L11-16, p25 L16-22).

The "cluster analysis" panels were derived from images of the mCherry channel subjected to thresholding and analyzed for particles larger than 500 pixels². This is now

explained in the Methods and Protocols section (p25 L22-p26 L2) as well as in the corresponding figure legends.

p.7 line 18. While unrelated species cluster with a *B. subtilis* pLS20 donor, was there any indication of SSB-YFP transfer to *B. cereus* or *B. megaterium*? I presume not but maybe pLS20 genes aren't expressed in those organisms. Whatever the case, I feel there should be mention of whether or not this happened and perhaps some explanation as to why or why not.

Although pLS20 is widespread in *Bacilli*, we were unable to monitor the transfer of pLS20 to *B. cereus* or *B. megaterium* using both the SSB-YFP reporter and CFU assays. These observations are in line with previous reports where direct transfer of pLS20 to these species was not detected, but pLS20-mediated transfer of other mobilizable plasmids was detected (Koehler & Thorne, 1987), highlighting the importance of pLS20-induced mating clusters in nature. This is now discussed in the revised manuscript (p16 L8-10).

P.7 line 21. To the best of my knowledge, a *flgM* mutant is not hyperflagellated in *B. subtilis* or other bacteria. Mutation of *FlgM* increases SigD/Sig28 dependent gene expression but these are late class flagellar genes. P. 8 line 18. I also don't believe that a *flgM* mutant has been reported to be hypermotile. While not quantitative, the assay in Fig 2D might suggest that the *flgM* mutant is more motile than the wild type but I'm not sure why or how that would be. Maybe *motAB* overexpression?

In *B. subtilis*, it has been reported that the influence of SigD is not limited to late-class flagellar genes, but also extends to the regulation of other flagellar components including flagellar hook, basal body, and export apparatus (Mukherjee & Kearns, 2014). In agreement, we now provide additional data suggesting that there are indeed more flagellated motile cells in the *flgM* mutant population in comparison to WT population, as supported by the motility assay in Fig. 2D. This data is now presented in the revised manuscript (Appendix Fig. S2B).

Fig 2EF. What is the Y-axis on these graphs? Is it linear, or log? It looks like neither. I believe the data was generated by the frequency of antibiotic resistance gene transfer so the dynamic range should be much larger than what is shown here. What was the lower limit of detection? 10^{-7} ? 10^{-8} ? On a log scale that goes to lower limit, the effects of mutants like *hag* or *sigD* could look massive, whereas the effects of mutants like *motA* or *flgM* might look to be barely detectable. In short, could these results be projected on a graph similar to that of Fig 3A?

As requested by the reviewer, the Y-axis of all the graphs, presenting CFU-based conjugation frequencies, have been modified to include a larger dynamic range (10^{-6} to 10^{-1}) (Fig. 2E-2F; Fig. 6B, 6J).

p. 18 line 3. "Our findings unravel the riddle of efficient conjugation of pLS20 in liquid medium despite lacking an apparent pilus" Clarify. I feel like the work aids our

understanding of why motility is required but I don't see how it explains the lack of pilus.

This statement has been modified accordingly in the revised manuscript (p18 L14-16).

References

- Couturier, A., Virolle, C., Goldlust, K., Berne-Dedieu, A., Reuter, A., Nolivos, S., Yamaichi, Y., Bigot, S., & Lesterlin, C. (2023). Real-time visualisation of the intracellular dynamics of conjugative plasmid transfer. *Nature Communications*, *14*(1), 294. <https://doi.org/10.1038/s41467-023-35978-3>
- Gago-Córdoba, C., Val-Calvo, J., Abia, D., Díaz-Talavera, A., Miguel-Arribas, A., Aguilar Suárez, R., van Dijl, J. M., Wu, L. J., & Meijer, W. J. J. (2021). A conserved class II type thioester domain-containing adhesin is required for efficient conjugation in *Bacillus subtilis*. *MBio*, *12*(2). <https://doi.org/10.1128/mBio.00104-21>
- Koehler, T. M., & Thorne, C. B. (1987). *Bacillus subtilis* (natto) plasmid pLS20 mediates interspecies plasmid transfer. *Journal of Bacteriology*, *169*(11), 5271–5278. <https://doi.org/10.1128/jb.169.11.5271-5278.1987>
- Masai, H., & Arai, K. (1997). Frpo: a novel single-stranded DNA promoter for transcription and for primer RNA synthesis of DNA replication. *Cell*, *89*(6), 897–907. [https://doi.org/10.1016/s0092-8674\(00\)80275-5](https://doi.org/10.1016/s0092-8674(00)80275-5)
- Mukherjee, S., & Kearns, D. B. (2014). The structure and regulation of flagella in *Bacillus subtilis*. *Annual Review of Genetics*, *48*(1), 319–340. <https://doi.org/10.1146/annurev-genet-120213-092406>

Dear Sigal,

Thank you for submitting a revised version of your manuscript. We have now received input from both original reviewers, who now find that their previous concerns have been addressed satisfactorily. Therefore, there now remain only a few editorial points that need addressing before I can extend official acceptance of the manuscript:

1. Please check that the funding information is correct and identical both in the manuscript and our online system. Currently, Golda Meir postdoctoral fellowship is acknowledged in the manuscript, but is missing in our online system.
2. CRedit has replaced the traditional author contributions section because it offers a systematic, machine-readable author contributions format that allows for more effective research assessment. Please remove the Authors Contributions from the manuscript and use the free text boxes beneath each contributing author's name in our online submission system to add specific details on the author's contribution. More information is available in our guide to authors.
3. Please move the "Data availability" section before "Acknowledgments".
4. Please rename "Methods and Protocols" section into "Methods".
5. Please remove EV table legends from the manuscript text file.
6. Our data editors have flagged the following issues in figure legends that need correcting:
 - Please provide the exact p values in the legends of figures 1d, f; 2e; 3a, c-d; 4e; 5f; 6j.
 - Please define the box plots in terms of minima, maxima, centre, bounds of box and whiskers, and percentile in the legend of figure 6j.
 - Please provide information on the nature and number of replicates in the legend of figure 6j.
 - Please define the error bars in the legends of figures 3a; 6b.
 - Please define the white arrows in the legend of figure 7b.
7. Papers published in The EMBO Journal are accompanied online by a 'Synopsis' to enhance discoverability of the manuscript. It consists of A) a short (1-2 sentences) summary of the findings and their significance, B) 3-4 bullet points highlighting key results and C) a synopsis image that is 550x300-600 pixels large (width x height, jpeg or png format). You can either show a model or key data in the synopsis image. Please note that the image size is rather small and that text needs to be readable at the final size. Please send us this information together with the revised manuscript.

With best wishes,

Ieva

We realize that it is difficult to revise to a specific deadline. In the interest of protecting the conceptual advance provided by the work, we recommend a revision within 3 months (23rd Jan 2025). Please discuss the revision progress ahead of this time with the editor if you require more time to complete the revisions.

Referee #1:

Bhattacharya, Bejerano-Sagie et al. substantially revised their manuscript and appropriately addressed my concerns. I

congratulate the authors on this impressive work and recommend publication of the manuscript.

Referee #2:

The authors have addressed my concerns.

The authors addressed the remaining editorial issues.

Dear Sigal,

Thank you for addressing the final editorial points. I am now pleased to inform you that your manuscript has been accepted for publication in the EMBO Journal.

Before we forward your manuscript to our publishers, I would like to propose some edits in the manuscript title, abstract and synopsis (please see below and the attached text file). I have also written a short blurb that will accompany the title of your manuscript in our online system. Please let me know if any corrections or adjustments are needed.

New title:

Flagellar rotation facilitates the transfer of a *Bacillus subtilis* conjugative plasmid

Blurb:

Sensing of flagellar rotation by the two-component system DegS-DegU induces expression and transfer of the pLS20 conjugative plasmid in a liquid environment.

Synopsis

Unlike typical bacterial conjugation systems that are activated upon bacterial contact with solid surfaces, the pLS20 plasmid, prevalent in Bacilli, is predominantly transferred in fluid environments that require motility. This study shows that pLS20 conjugation gene expression is activated by a host mechanosensing pathway that discerns flagella rotation, a strategy enabling plasmid invasion into remote habitats.

- Conjugation mediated by pLS20, carried by *B. subtilis*, is motility dependent.
- Formation of mating clusters allows conjugation in fluid surroundings.
- Conjugation gene expression is induced by flagella rotation sensing by the DegS-DegU two-component system.
- Synchronizing conjugation with motility promotes plasmid spread into new ecological niches.

If you have any questions, please do not hesitate to contact the Editorial Office. Thank you for your contribution to The EMBO Journal, and congratulations with a nice study!

Best wishes,

Ieva
